# Assessing quantitative MRI techniques using multimodal comparisons

**Francis Carter** [1,2], **Alfred Anwander**[3], **Mathieu Johnson**[1], **Thomás Goucha**[3], **Helyne Adamson**[3], **Angela D. Friederici**[3], **Antoine Lutti**[4], **Claudine J. Gauthier**[5,6,7], **Nikolaus Weiskopf**[8,9], **Pierre-Louis Bazin**[10,11], **Christopher J. Steele**[1,6,11]*

**1** Department of Psychology, Concordia University, Montreal, Québec, Canada, **2** Montreal Institute for Learning Algorithms, Université de Montréal, Montreal, Québec, Canada, **3** Department of Neuropsychology, Max Planck Institute for Human Cognitive and Brain Sciences, Leipzig, Germany, **4** Department of Clinical Neurosciences, Lausanne University Hospital and University of Lausanne, Lausanne, Switzerland, **5** Department of Physics, Concordia University, Montreal, Québec, Canada, **6** School of Health, Concordia University, Montreal, Québec, Canada, **7** Montreal Heart Institute, Montreal, Québec, Canada, **8** Department of Neurophysics, Max Planck Institute for Human Cognitive and Brain Sciences, Leipzig, Germany, **9** Felix Bloch Institute for Solid State Physics, Faculty of Physics and Earth Sciences, Leipzig University, Leipzig, Germany, **10** Faculty of Social and Behavioral Sciences, University of Amsterdam, Amsterdam, Netherlands, **11** Department of Neurology, Max Planck Institute for Human Cognitive and Brain Sciences, Leipzig, Germany

* christopher.steele@concordia.ca

## Abstract

The study of brain structure and change in neuroscience is commonly conducted using macroscopic morphological measures of the brain such as regional volume or cortical thickness, providing little insight into the microstructure and physiology of the brain. In contrast, quantitative Magnetic Resonance Imaging (MRI) allows the monitoring of microscopic brain change non-invasively in-vivo, and provides directly comparable values between tissues, regions, and individuals. To support the development and common use of qMRI for cognitive neuroscience, we analysed a set of qMRI and dMRI metrics (R1, R2*, Magnetization Transfer saturation, Proton Density saturation, Fractional Anisotropy, Mean Diffusivity) in 101 healthy young adults. Here we provide a comprehensive descriptive analysis of these metrics and their linear relationships to each other in grey and white matter to develop a more complete understanding of the relationship to tissue microstructure. Furthermore, we provide evidence that combinations of metrics may uncover informative gradients across the brain by showing that lower variance components of PCA may be used to identify cortical gradients otherwise hidden within individual metrics. We discuss these results within the context of microstructural and physiological neuroscience research.

## Introduction

Magnetic Resonance Imaging (MRI) gives rise to a variety of image contrasts primarily driven by differences in longitudinal and transverse relaxation rates of the

**Data availability statement:** Data cannot be shared publicly because data sharing was not allowed by the ethics committee. The data underlying the results presented in the study are available from the senior author or by contacting the University of Leipzig Ethics Advisory Board using the information at the following link: https://www.uni-leipzig.de/en/research/integrity-of-research/ethics-advisory-board.

**Funding:** CJS was supported by the Natural Sciences and Engineering Research Council (NSERC: RGPIN-2020-06812, DGECR-2020-00146) and the Heart and Stroke Foundation of Canada New Investigator Award and Catalyst from the Canadian Institutes of Health Research (HNC 170723). PLB was supported by the NWO Vici grant (PI: Birte Forstmann). https://www.nwo.nl/en/researchprogrammes/nwo-talent-programme CJG was supported by the Heart and Stroke Foundation of Canada New Investigator Award, Michal and Renata Hornstein Chair in Cardiovascular Imaging, and NSERC (DG: RGPIN-2015-04665). AA and AF received funding from the SPP2041 program "Computational Connectomics" of the German Research Foundation (DFG). NW has received funding from the European Research Council under the European Union's Seventh Framework Programme (FP7/2007-2013) / ERC grant agreement no. 616905; from the European Union's Horizon 2020 research and innovation programme under the grant agreement No 681094; from the BMBF (01EW1711A & B) in the framework of ERA-NET NEURON.

**Competing interests:** The Max Planck Institute for Human Cognitive and Brain Sciences has an institutional research agreement with Siemens Healthcare. NW holds a patent on acquisition of MRI data during spoiler gradients (US 10,401,453 B2). NW was a speaker at an event organized by Siemens Healthcare and was reimbursed for the travel expenses.

MRI signal between voxels. The estimates of these relaxation rates, provided by quantitative MRI (qMRI) [1,2], have been shown to correlate with microstructural properties of brain tissues such as iron or myelin concentration [3,4] and show great potential for the monitoring of microscopic brain change in-vivo [5,6]. However, the specificity of these biomarkers is limited by the combined contributions of multiple histological properties to qMRI estimates [2,5–8]. Improving their specificity requires an understanding of how individual tissue properties are reflected in qMRI estimates. Towards this goal, the current study quantifies the specificity and overlap of the information contained within a set of quantitative MRI metrics to assist in the development of "in-vivo histology", the mapping of MRI signals to microstructural tissue properties [2,6].

The assessment of microscopic brain tissue properties from in-vivo qMRI data requires a detailed understanding of how these properties impact the MRI signal [2,6]. Recent research includes evidence that the MR relaxation rates R1 and R2* are related to myelin and iron concentrations [6,9–13] and that magnetization transfer (MT/MTSat/MTR) measurements scale with myelin's contribution to the macromolecular content [14–16]. Myelin concentration has been shown to be the dominant contributor to multiple qMRI measures [17,18], highlighting the need for complementary qMRI measures in order to achieve a complete description of brain microstructure.

One major issue is that the MR signal from a single voxel is potentially the result of many different molecular arrangements and concentrations, and therefore provides ambiguous information about the microstructural composition in that voxel [5,7,8]. Multimodal MRI – where multiple MRI contrasts are acquired in the same participant or sample – may help address this issue. Multiple qMRI measures with different relationships to tissue properties can be combined to extract latent variables that capture shared variance. Extracted latent variables can be related to microstructural features and ground-truth molecular concentrations to determine how well they map to specific tissue properties, such as disentangling microstructural differences related to diffusion hinderance from those related to orientation and fiber organization [19–22] However, this requires accurate segmentation and subsequent reduction of tissue partial voluming, which has a large but often overlooked effect on the interpretation of results [23], and is often difficult to properly implement for cortical grey matter where partial voluming with dura/cerebrospinal fluid (CSF) and white matter (WM) is common at standard resolutions [24–26]. qMRI has other advantages, including higher reproducibility and comparability between acquisitions within and between participants than conventional structural MRI commonly used for assessing macroscopic morphological change, as well as diffusion MRI commonly used for assessing microstructural integrity [6,27,28]. The advent of more advanced qMRI sequences that simultaneously capture multiple quantitative metrics has led to shorter overall acquisition times [29,30] and removes the need for within-subject registration between metrics. Notably, this work has led to the development of Multi-Parametric Mapping, which simultaneously captures quantitative R1, R2*, MTSat, and PD images [MPM; 30] and multi-echo MP2RAGE [T1, T2*, QSM; 27,31].

While useful for acquiring multiple co-registered modalities, and often more robust in explaining variation in brain microstructure [32], multiparametric methods likely contain redundancies in the information provided by each metric [2,6,33], especially given the fact that dMRI and qMRI metrics show high correlation with each other [34]. In fact, in certain circumstances, a subset of qMRI and dMRI metrics that show redundancies to one other metrics present in the analysis can be removed without significantly reducing the variance explained by the remaining metrics [20]. As such, these redundancies have the potential to be exploited to determine which combination of metrics accounts for the most variance in the tissue of interest (and could therefore form a basis for a minimum useful multiparametric acquisition profile) and, importantly, can then be used to map putative microstructural similarities across the brain [6]. Similar to the way that microstructure exhibits differences across different tissues and regions, the relationships between metrics are also expected to vary. Previous work has explored metric-metric covariance relationships with ROI-based or segmented network approaches [35,36] such as analyzing metric covariance in predetermined WM tracts of interest [20,22,37], and a within-ROI binning approach [21], which rely on *a priori* regional delineations and averaging that may obscure subtle variability within regions. A voxel-wise approach could provide a more nuanced and comprehensive description of these relationships and help to determine if and/or how multiparametric combinations can provide additional information not found in individual metrics [38].

The present study provides a statistical description of the linear relationships for a set of six metrics, including four qMRI metrics from the MPM [30] (R1, R2*, Magnetization Transfer saturation – MTSat, and Proton Density saturation – PD) and two diffusion MRI metrics (dMRI: Fractional Anisotropy – FA, and Mean Diffusivity – MD). We set out to identify potential redundancies between metrics, and additionally identify the set of normative relationships between metrics across different tissue types (sub/cortical grey matter, white matter) that can be used to develop and test hypotheses in the development of in-vivo histology. We also provide evidence that multimodal imaging holds promise in describing the microstructural drivers of qMRI contrast by applying PCA to extract and map linear latent variables within the data [22].

## Methods

### 2.1. Participants

Our sample consisted of 101 young healthy individuals (mean age 25.7±4, range 18–34 years, 25 females) who were recruited in the area of Leipzig, Germany. All participants were right-handed, had a high-school-level education, no history of neurological or psychiatric disorders, and did not use any centrally effective medication. The study was approved by the ethics committee at the medical faculty of the University of Leipzig. All participants gave written informed consent and were paid for participation in the study.

### 2.2. MRI data acquisition and quantitative metric estimation

Quantitative multi-parametric maps (MPM, 1 mm iso) and high spatial and angular resolution diffusion-weighted MRI (dMRI, 1.3 mm iso) were acquired for all participants on a 3-Tesla PRISMAfit MRI system (Siemens Healthineers, Erlangen, Germany) using a standard 32-channel receive head coil. The MPM protocol consisted of three multi-echo 3D FLASH (fast low angle shot) acquisitions with predominant T1-, PD-, and MT-weighting by adapting the repetition time (TR) and the flip angle (T1w: 18.7 ms/20°, PDw and MTw: 23.7 ms/6°) following the setting previously published [30]. The MTw acquisition included an additional initial 4 ms long off-resonance MT saturation pulse (flip angle 220°, 2 kHz offset) [39]. We acquired six gradient echoes (equidistant echo times, TE, from 2.2 to 14.7 ms) with alternating polarity for the T1w and the MTw volumes and eight datasets for the PDw volumes (TE from 2.2 to 19.7ms). The remaining imaging parameters were: field-of-view 256×240 mm, 176 slices, acceleration in phase-direction using GRAPPA 2 and in partition direction using 6/8 partial Fourier, non-selective RF excitation, high readout bandwidth=425 Hz/pixel, RF spoiling phase increment=50°, acquisition time ~21min. The acquisition was preceded by additional calibration data to correct for spatial

variations in the radio-frequency transmit field (B1+) as described by Lutti and colleagues [40]. The diffusion MRI protocol used the multi-band sequence developed at CMRR (https://www.cmrr.umn.edu/multiband) with the following parameters: 1.3 mm isotropic resolution, b-value = 1000 s/mm$^2$, 60 directions and seven images without diffusion weighting (b = 0), three repetitions to improve the SNR, TE = 75 ms, TR = 6 s, GRAPPA 2, CMRR-SMS 2, two b = 0 acquisitions with opposite phase encoding directions. The acquisition time for the dMRI protocol was 23 min.

The MPMs acquisitions were processed with the freely available hMRI toolbox (http://hmri.info, Tabelow et al., 2019) to estimate parameter maps of the longitudinal relaxation rate (R1), the effective transverse relaxation rate (R2*), the magnetization transfer saturation (MTSat), and the proton density (PD) for each participant. The dMRI images were corrected for motion, eddy currents, and susceptibility distortions using the FSL TopUp toolkit [41], and the fractional anisotropy (FA), and mean diffusivity (MD) were computed from the diffusion tensor images using the FSL tools [42]. Registration from native diffusion to higher resolution MPM space was conducted using FSL FLIRT and was accomplished by rigidly co-registering the b0 and R1 images within each individual, and applying the transform to FA and MD maps to bring them into R1 space (trilinear interpolation).

## 2.3. Preprocessing

After data collection and quantitative metric reconstruction, the following steps were performed to segment the brains into different tissue types (cerebral cortical GM, subcortical GM, WM), and to create surface representations of the cortex. We obtained high-quality segmentations and cortical representations by employing previously established methods [43–45]. Other than the SPM skull stripping and segmentations described below, all other preprocessing was performed with the open-source neuroimaging toolbox Nighres v1.2 [46].

**2.3.1. Segmentation.** Cortical, subcortical, and WM segmentations for each participant were performed with standard GM segmentation in SPM12 on the R1 image [47] and the Nighres implementation of Multiple object Geometric Deformable Model (MGDM) segmentation using FA, R1, MTSat, and PD [43,48]. We utilized the multiple segmentations in combination with rigorously identified threshold cutoff values to restrict our analyses to only the tissues of interest (cortex, subcortex, WM) while excluding voxels that included partial voluming (e.g., GM that includes WM, CSF, or dura) or artifactual signal. Final masks were inspected for each individual by F.C. to ensure that they corresponded only with the specific tissue type (cortex, subcortex, WM). MGDM allowed for the segmentation of the cerebral cortical and subcortical GM while preserving topology, and SPM segmentations were combined as follows to ensure final segmentations that reduced tissue partial voluming. WM masks were defined as the intersection between SPM WM probability maps thresholded at 0.5 and the MGDM cerebral WM. The MGDM provides a topologically correct tissue segmentation that is generally more conservative than SPM's WM segmentation and their intersection ensured that any partial voluming effects from either approach would not be included in the final mask. GM masks were defined as the intersection between the SPM GM (probability maps thresholded at 0.95) and the MGDM labels for cerebral GM. We used a more conservative threshold for the SPM GM probability maps in this step to eliminate non-brain tissue at the pial surface to allow for a more accurate cortical extraction (see 2.3.2). As for cortical grey matter, subcortical GM masks were defined as the intersection between the MGDM subcortical labels and the SPM GM thresholded at 0.95. The resulting masks retained artifactual R2* signal (mainly in the medial temporal lobe) which we removed by excluding voxels containing R2* values above 25 s$^{-1}$ in GM (WM was not thresholded). We also used an MD threshold of 0.001 mm$^2$s$^{-1}$ x $10^{-6}$ to reduce any partial voluming with CSF in the GM masks. These threshold values were chosen with reference to the anatomical images and segmented tissue masks in each individual and the same values used across the entire group to ensure comparable tissue type identification. All masking and metric thresholds were chosen to restrict our analyses as closely as possible to tissue type (GM, WM) and optimally reduce partial voluming and signal artifacts. Subsequent analyses were performed on voxel-wise values extracted from the cortical sheet, WM, and subcortical GM in both hemispheres. We also repeated these analyses without any metric thresholds and found very similar results with identical significant effects (data not shown). A group

surface co-registration step for both hemispheres was additionally used to project cortical results onto a common group surface for visualization as detailed below.

   **2.3.2. Cortex extraction, mesh generation, and surface projection.** A 3D mesh representation of the cortex was used to map and visualize our results on the cortical surface. To mitigate against the effects of partial voluming on the pial and WM surfaces, we extracted a thin sheath of the middle layer of the cortex with Nighres as follows. The cortex was first extracted with CRUISE Cortex Extraction [26], generating GM-WM and GM-CSF boundary levelset images (where a levelset is the distance of each voxel from each boundary; thus representing the inner and outer cortical surfaces in voxel space). These two boundaries were then combined to extract a levelset representation of the midline of the cortical sheet. The midline levelset, used later for group co-registration, was then thresholded at a distance of 0.5 mm to generate a binary mask, and converted to a 3D mesh for display. Details on segmentation and the levelset approach for the surface generation with Nighres can be found in our previous technical publications [46], and all specific commands and parameters are included as code in our github repository (https://github.com/neuralabc/paper_QuantitativeMetricComparisons).

## 2.4. Group co-registration

Cortical midline levelsets were used to perform surface-based registration in voxel space with a method similar to that developed by Tardif and colleagues [45]. The ANTs [49,50] multivariate template construction script was used to perform nonlinear registrations and generate a group template for both hemispheres. The levelsets representing each participant's middle cortical ribbon were thresholded at 10 mm from the cortex and used as inputs to ANTs Multivariate Template Construction (v2, 3 iterative steps, with default linear and nonlinear registration parameters, Demons similarity metric). The final template levelset was then converted to a mesh representation for display [46]. To map the results of the analysis onto a mesh representation of the group template surface, participants' data from each metric was: 1) multiplied by a binarized midline ribbon, 2) nonlinearly transformed into the study-specific group space with the computed deformation (ANTs SyN, nearest-neighbor interpolation), and 3) a 2 mm intensity propagation was then performed to project participants data onto the final coregistered surface. In all cases, the vertex-wise median values across participants were presented for display.

## 2.5. Statistical analysis

   **2.5.1. Metric-metric density plots.** Bivariate logarithmic density plots were created to describe the relationship between metrics in GM, WM, cortex, and subcortical GM. Linear regressions were used to calculate the coefficients of determination (R-squared) for each bivariate relationship. Univariate density plots (histograms) were also generated to further describe the data.

   **2.5.2. Metric-metric correlation tissue type comparisons.** We sought to identify potential differences between tissue types on their metric-metric relationships as an initial exploration for how similar these relationships are across tissues. We computed the metric-metric correlations for each individual and tissue type and then used paired t-tests to compare correlation values between each tissue type. For example, for FA vs MD each participant's FA-MD in WM and FA-MD in cortical GM corelation r-values were entered into a paired t-test to compare this relationship between WM and cortical GM. All comparisons were Bonferroni corrected for multiple comparisons. While age and sex differences have been shown to have an impact on brain structure [51,52], this analysis did not include these variables as covariates. In our analyses we aimed to identify the linear pair-wise relationships between metrics, which should be relatively robust to these differences, rather than assess individual quantitative metrics. Furthermore, many of the metrics used here sample from a subset of physiological markers, hence their redundancies. This suggests that the differences due to age and sex would be distributed across the metrics of interest, which would limit their impact on the pairwise correlations we computed. In future research with a larger, sex-balanced, sample that spans a larger age range, subtle differences in these physiological factors could be explored.

**2.5.3. Dimensionality reduction.** As an additional exploratory step, we performed model-free dimensionality reduction with Principal Component Analysis (PCA) in cortical tissue to visualize the latent variables that drive the variance within the data. All six metrics (R1, R2*, MTSat, PD, FA, and MD) in each of these tissues from all participants were z-scored and used as input variables to a PCA in scikit learn [53] in Python (3.7), and all six PCs were assessed. Each participant's data was then transformed into the PCs' subspace for visualization, and transformed into the common group space. This allowed us to compute the median across participants and visualize our results on the cortical surface template.

## Results

### 3.1. Cortical segmentation

Our segmentations combined MGDM and SPM segmentations to reduce biases caused by partial voluming of GM with WM or CSF. Visual inspection showed that our results had limited partial voluming of the cortex with CSF and WM, with little to no gyral or sulcal bias [25; Fig 1]. As an additional confirmatory step before using this segmentation in our analyses, we also checked to ensure that our cortical maps showed high correspondence with those presented in previous literature. As expected, R1 cortical maps exhibited a gradient with peaks in primary sensory and motor regions (Fig 1), a pattern that is in correspondence with what has been shown in previous literature [11,54]. Surface plots for all six metrics can be found in S1 Fig.

### 3.2. Descriptive statistics: GM and WM

All pairs of metrics, both in GM (combined cortical and subcortical) and WM, were found to be significantly correlated (Fig 2, all $p < 0.001$). However, there were large fluctuations in these pairwise correlations, with $R^2$ values ranging from as low as 0.0001 in the MD-R2* GM correlation, to as high as 0.482 in the R1-MTSat GM correlation. These differences were not equally dispersed across metrics; R1 had the highest correlations with other metrics across all structures ($R^2$ average = 0.258), while MD had the weakest correlations ($R^2$ average = 0.051). Furthermore, the strengths of correlations varied across tissues, with some $R^2$ values differing by over 10% of explained variance. Notably, FA-MD, MD-MTSat, and R1-MD all had $R^2$ values that were more than 10% higher in GM than in WM, and almost all pairwise correlations were greater in GM than WM. However, FA-PD and FA-R2* had $R^2$ values slightly higher in WM than in GM. These relationships were reliable and consistent across participants, as shown by the low standard deviation of $R^2$ across participants in S2 Fig.

**3.2.1. Descriptive statistics: WM and subcortical GM.** All pairs of metrics, both in subcortical GM and WM, were found to be significantly correlated (S6 Fig). However, there were large fluctuations in these pairwise correlations, with R2 values ranging from as low as <0.005 in the FA-MD and FA-R2* subcortical correlations, to as high as 0.4465 in the PD-R1 subcortical correlation and 0.4198 in the PD-R1 WM correlation. These differences were not equally dispersed across metrics; R1 had the highest correlations with other metrics across all structures (R2 average = 0.223), while MD had the weakest correlations (R2 average = 0.027). Furthermore, the strengths of correlations varied across tissues, with some R2 values differing by over 10% of explained variance. Notably, R1-R2* and MTSat-FA correlations explained 11.3 and 9.88% more variance in cortical GM than in WM. These relationships were reliable and consistent across participants, as shown by the low standard deviation of R2 across participants in S4 Fig.

**3.2.2. Descriptive statistics: WM and cortical GM.** All pairs of metrics, both in cortical GM and WM, were found to be significantly correlated (S7 Fig). However, there were large fluctuations in these pairwise correlations, with R2 values ranging from as low as <0.005 in the PD-R2* cortical correlation, to as high as 0.428 in the R1-MTSat cortical correlation. These differences were not equally dispersed across metrics; R1 had the highest correlations with other metrics across all structures (R2 average = 0.218), while MD had the weakest correlations (R2 average = 0.046). Furthermore, the strengths

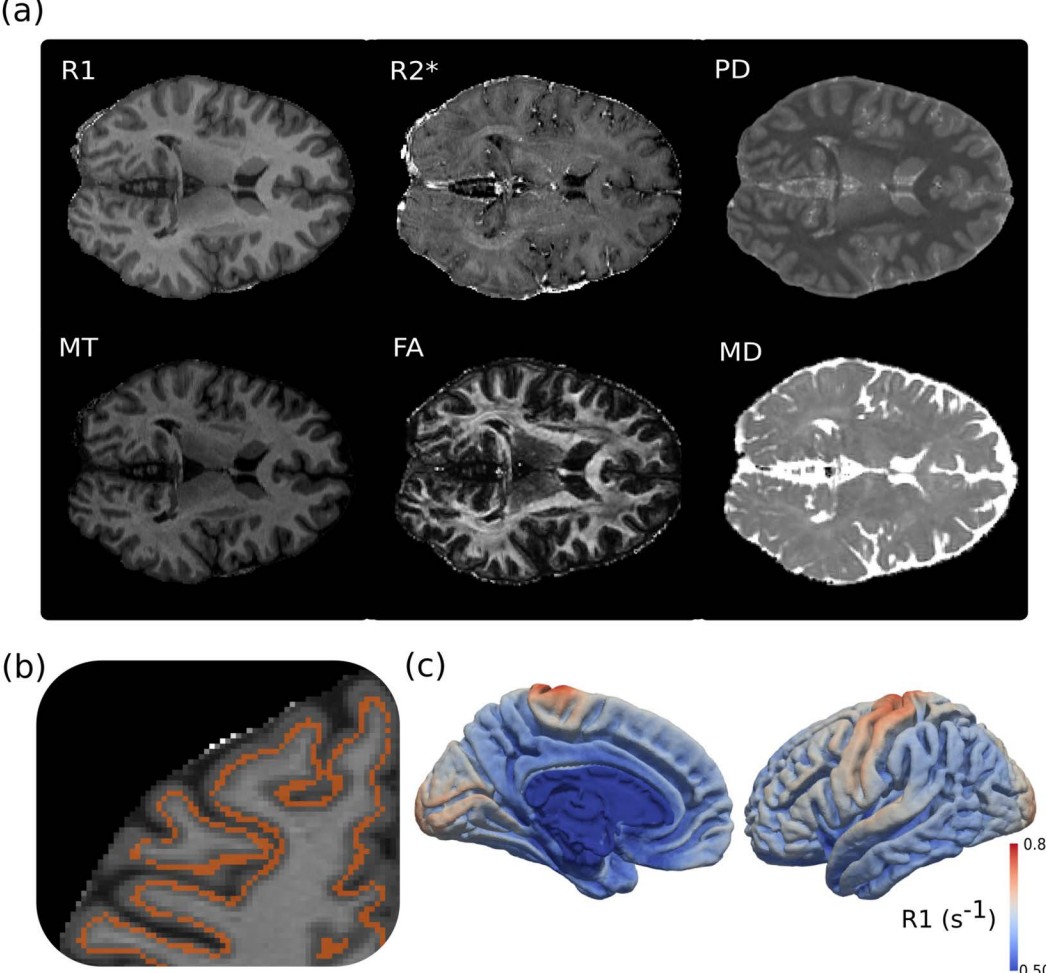

(a)

(b) (c)

R1 (s$^{-1}$)

**Fig 1. Overview of segmentation results.** (a) The six metrics used were captured in 101 participants. (b) A sample brain with a red mask of the middle cortical ribbon as a semi-transparent overlay on R1. (c) Using data from the cortical ribbon, we plotted the median R1 values across all participants to the group template. The gradient here corresponds to the R1 values mapped back onto the brain, and shows overall higher values in the sensory and motor areas of the cortex that decrease with cortical distance. Other metric plots can be found in S1 Fig.

of correlations varied across tissues, with some R2 values differing by over 10% of explained variance. Notably, FA-MD and MTSat-MD correlations explained 21.5 and 9.75% more variance in cortical GM than in WM, however. The R1-PD correlation explained 9.18% more variance in WM than in cortical GM. These relationships were reliable and consistent across participants, as shown by the low standard deviation of R2 across participants in S5 Fig.

**3.2.3. Metric-metric correlation tissue type comparisons.** Overall, the pairwise correlations were most similar when comparing cortical and subcortical GM, and most different when comparing cortical GM with WM. As a summary estimate of the difference between metric-metric relationships across tissue types, the mean absolute value of the t-values across all comparisons was 14.23 for cortical vs WM, 12.62 for subcortical vs. WM, and 12.25 for cortical vs. subcortical. Across all comparisons, there were only a few relationships that were not significantly different after multiple comparisons correction. Cortical vs WM: R2/MD ($t=-1.09$, $p=1$). Subcortical vs WM: R1/MTSat ($t=-0.81$, $p=1$), R2/PD ($t=1.83$, $p=1$). Cortical vs subcortical: R1/MD ($t=2.28$, $p=1$).

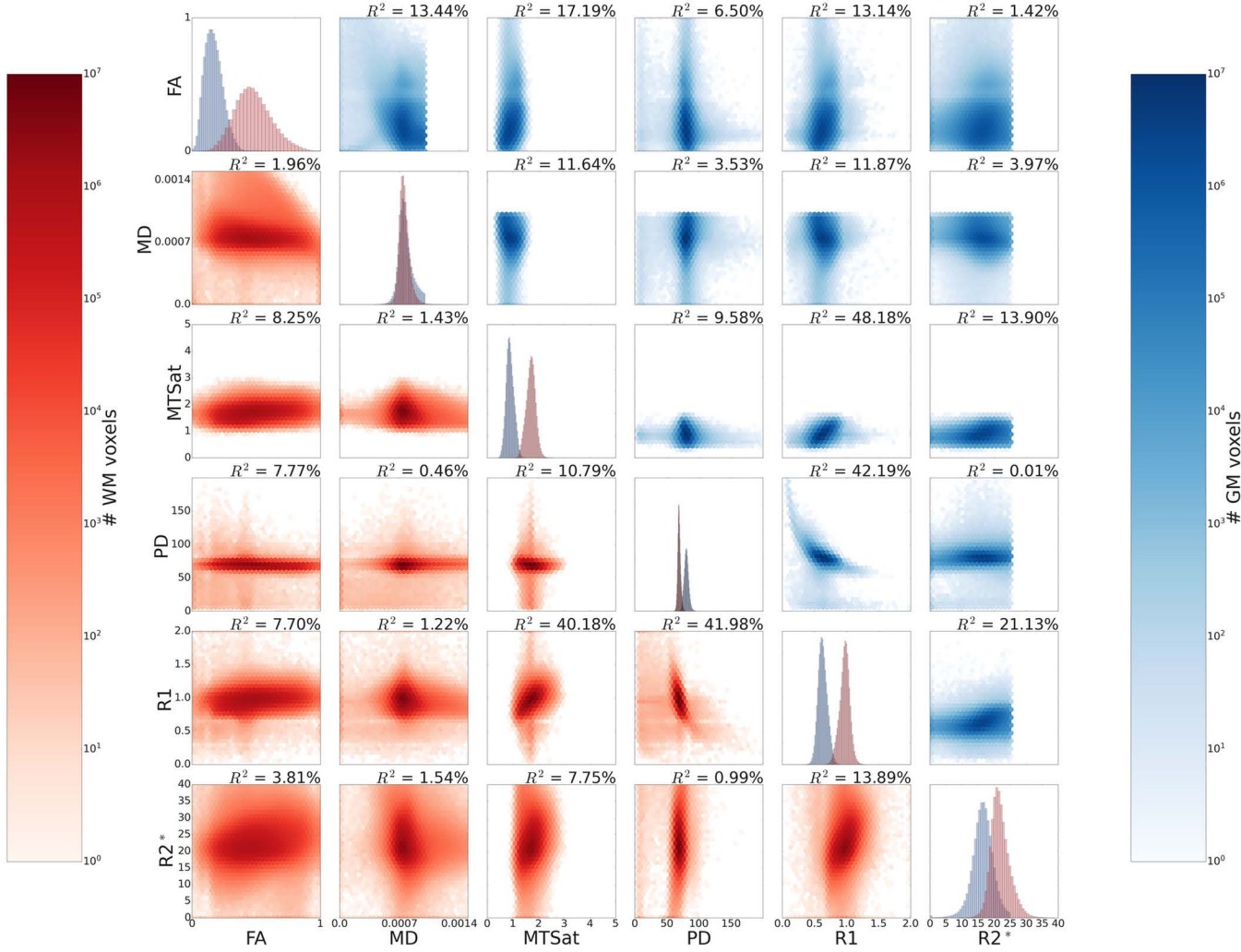

**Fig 2. Pairwise correlations in GM (above the diagonal) and WM (below the diagonal) for all participants' data.** P-values for all pairwise correlations were very small (<0.001), however were not stated explicitly as the small values of p do not necessarily relate to higher explained variance. $R^2$ values for each pairwise correlation are shown above their respective graphs. Data along the diagonal shows the histograms for each metric in GM (blue) and WM (red). GM includes values extracted from the midline cortical ribbon and subcortical nuclei. Metric thresholds were implemented for R2* and MD to reduce partial volume effects and artifacts, and are visible as sharp cutoffs in the figure (see Method). Pairwise density plots are depicted in log space to show the full range of data.

### 3.3. Descriptive statistics: Cortical and subcortical GM

All pairs of metrics were found to be significantly correlated in cortical and subcortical GM (Fig 3, all p<0.001). However, there were large fluctuations in these pairwise correlations, with $R^2$ values ranging from as low as <0.005 in the PD-R2* cortical correlation, to as high as 0.447 in the R1-PD subcortical correlation. As with the GM and WM comparisons (Fig 2), differences were not equally dispersed across metrics; R1 had the highest correlations in all GM structures ($R^2$ average = 0.230), while MD had the weakest correlations ($R^2$ average = 0.071). Furthermore, the strengths of correlations varied across tissues, with some $R^2$ values differing by over 10% of explained variance. Notably, FA-MD and FA-R1 correlations explained 23.0 and 9.1% more variance in cortical GM than in subcortical GM, respectively. Conversely, the R1-PD and R1-R2* relationships

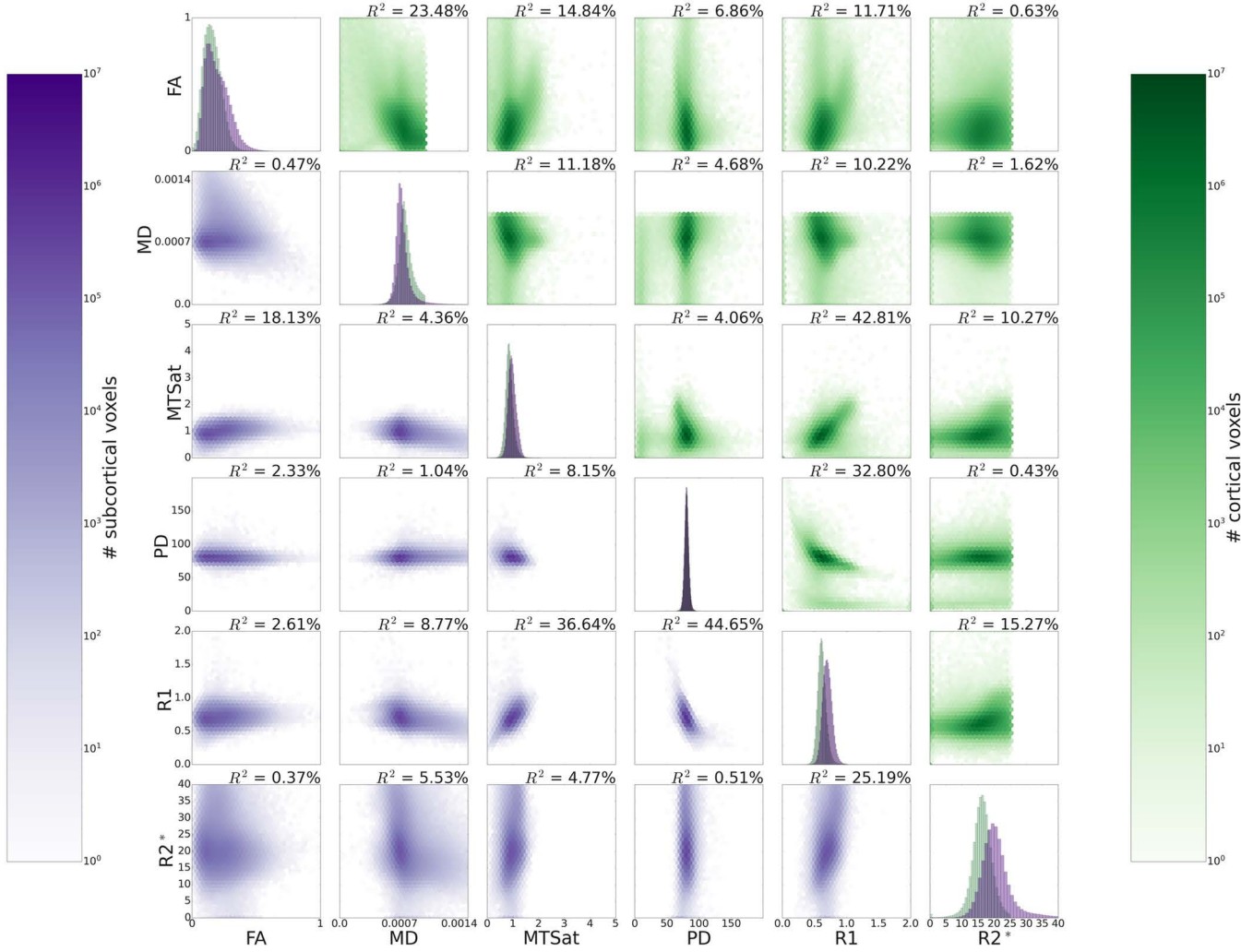

**Fig 3. Pairwise correlations in cortical GM (above the diagonal) and subcortical GM (below the diagonal) for all participants' data.** P-values for all pairwise correlations were very small (<0.001), however were not stated explicitly as the small values of p do not necessarily relate to higher explained variance. $R^2$ values for each pairwise correlation are shown above their respective graphs Data along the diagonal shows the histograms for each metric in cortical (green) and subcortical GM (purple). Metric thresholds were implemented for R2* and MD to reduce partial volume effects and artifacts, and are visible as sharp cutoffs in the figure (see Method). Pairwise density plots are depicted in log space to show the full range of data.

were 11.2 and 9.1% stronger in subcortical GM than cortical GM. These relationships were reliable and consistent across participants, as shown by the low standard deviation of $R^2$ across participants in S4 Fig.

### 3.4. Multimodal data visualization through dimensionality reduction

The results of the PCA revealed various patterns of cortical gradients, i.e., the derived values of the principal components mapped back onto the surface of the cortex for visualization. These analyses were done on all participants, but were also stable when applied to individuals, as suggested by the low variance in correlations across participants (S2–S5 Figs). All PCA components, when mapped back onto the cortex, provided informative gradients, often denoting primary sensory and motor areas (Fig 4). Interestingly, the components that explained the most variance in the metrics seemed

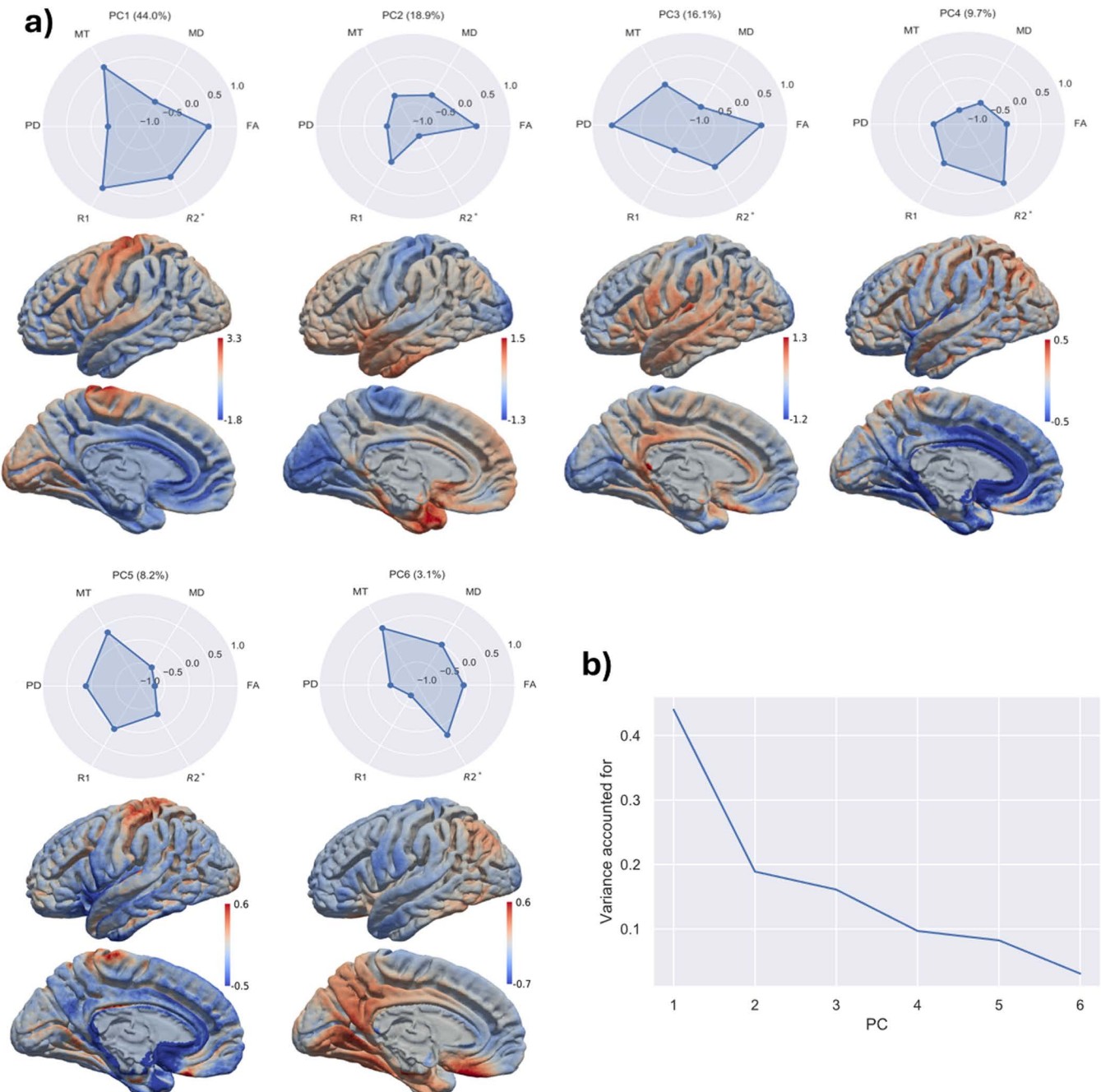

**Fig 4. Results of the PCA.** (a) radar plots showing the weighting of each component (ranging from −1 to 1) on each PC. In brackets is the percent variance accounted for by individual PCs. Below the radar plots, each PC has been mapped back onto the brain, showing the brain regions in which the variance was most explained by the each PC. Of particular note are PC1, showing primary sensory and motor regions and PC6. Although PC6 only accounts for 3.1% of the variance in the data, this component shows an interesting cortical gradient. **(b)** A plot of variance accounted for in each PC.

to diverge the least from the gradients of the metrics themselves. For example, PC1, which explains the most variance (44.0%), had a high weighting on R1 and metrics with strong correlations with R1. PC1's cortical gradient showed high correspondence with primary sensory and motor regions, similar to the R1 cortical map (Fig 1), as did the gradient of PC5,

albeit to a lesser degree. Similarly, PC2 resembled an inverse mapping of R2* and FA gradients, while PC3 resembled PD. The most novel gradient was in the component that explained the least amount of variance in contrasts (PC6, 3.1%) (Fig 4). This gradient least resembled individual quantitative metrics and showed a structural pattern having a degree of resemblance with the Default Mode Network, with high values in the posterior cingulate cortex, medial PFC, and posterior parietal cortex [44,55].

## Discussion

The goal of this study was to assess a set of quantitative metrics to provide a statistical description of their relationships. We used a set of previously established methods for extracting and co-registering the cortical ribbon [43,45] to identify core GM voxels that were less likely to suffer from partial volume effects. Bivariate correlations between metrics were performed for each tissue type, and our results showed that quantitative metrics have wide variations in their degree of correlation with each other, but these correlations are consistent across participants. In other words, different pairs of metrics exhibited varied relationships while the distribution of correlation values (and R2) across the group was highly consistent on any given pair of metrics. Furthermore, we found that applying PCA to cortical gray matter data resulted in the extraction of latent space components not obvious in individual unimodal metrics in the component accounting for the least variance.

Firstly, we observed the strongest overall pairwise correlation between R1 and MTSat, and the strength of this correlation was highly stable across tissue types ($R^2 = {\sim}36$–48%). This may be due to the strong link between R1 and myelin [6,10,13] and MTSat as a measure of macromolecular tissue content [16,21,56,57]; both of which are expected to be highly related throughout the brain [17,58]. The normative R1-MTSat relationships described here could potentially be affected by pathological conditions that affect macromolecular proportions, such as Alzheimer's tau pathologies and white matter hyperintensities, and baseline relationships would then be useful for research in these domains [59–61]. Repeating this analysis on a sample with various neurodegenerative diseases would increase the generalizability of results, and perhaps give us insight on how these diseases affect the brain if the principal components that drive the variance show differential patterns across pathologies.

Conversely, R2* had, on average, low correlations with other metrics ($R^2$ average ${\sim}=7\%$), indicating little redundancy. Its strongest correlation was with R1 (GM $R^2{\sim}=21\%$, WM $R^2{\sim}=14\%$) which is consistent with previous research indicating a link between R2* and myeloarchitecture [12,14], and an indirect link to myeloarchitecture through its relation with tissue iron concentration [6,9,10,13]. Interestingly, the R1-R2* correlation was weaker in WM where we might expect it to be highest based on their shared link with myeloarchitecture. Both these metrics are linked to myelin and iron concentrations [16,54,62], and so it would follow that they would have higher correlation than what we observed. One potential reason for this is that R2* is known to be affected by white matter fiber orientation relative to the main magnetic field (B0) [63,64]. This orientation-dependent effect could lead to variation in R2* that does not arise from myeloarchitecture or iron and therefore decreases the strength of the correlation within the white matter. This effect is not expected to be significant within grey matter since there is no predominant net orientation of fibers at voxel resolutions. Overall, the differences in R1 and R2* correlations provide further support for the hypothesis that R1 is more strongly related to other aspects of microstructure, such as the cyto- and myeloarchitecture [65] than R2*. Interestingly, the greater R1-R2* correlation in GM was predominantly driven by subcortical GM. Subcortical nuclei are relatively rich in iron [66] and we collapsed across all identified subcortical GM structures (caudate, putamen, thalamus, and globus pallidus), which may lead to the greater variability seen in the subcortex than cortex for both metrics (as indicated by the spread in histograms along the diagonal of Fig 3). This greater variability may in turn drive the greater R1-R2* correlation by making it easier to identify a statistical correlation. This leads us to expect that changes in iron distribution throughout the brain and particularly in the basal ganglia, as is often seen in neurodegeneration and related pathologies [67–69], may exhibit different R1-R2* correlations than what we have observed. Future work could seek to link the relationship between R1-R2* with other biomarkers of myelination and iron in patients diagnosed with pathologies related to iron accumulation for use as a non-invasive marker.

dMRI metrics (FA, MD) showed higher correlations with other metrics in GM than WM, as well as higher correlations in cortical GM than subcortical GM. This suggests that dMRI metrics may be dependent on tissue parameters that vary more independently in WM such as fiber orientation and directional uniformity, as would be expected from the diffusion tensor [70]. This then suggests that dMRI metrics in GM are more strongly driven by non-axonal microstructural differences than in white matter. For example, it is possible that differences in axonal radius and volume, as well as cell types and shapes could be a source of this change in dMRI correlations between the cortical and subcortical GM [71]. While FA differences/changes have previously been inferred to be due to differences in myelination, fiber orientation, and/or density [72–74], we found that there were low correlations between FA and R1 (GM $R^2 \sim 13\%$; WM $R^2 \sim 8\%$; cortical $R^2 \sim 12\%$; subcortical $R^2 \sim 3\%$), which is arguably a more direct indicator of myelin [13]. Past research has found that R1 and diffusion have differential variation throughout the lifespan, further supporting a difference in the information captured by these metrics, and suggesting that varying R1-dMRI correlations may be informative about health and ageing related issues [75]. We would also expect that the age-related changes in dMRI metrics would be region-specific, as aging affects myelination differentially across regions. It would therefore be beneficial to perform a similar analysis across developmental timepoints in addition to the present single-timepoint whole brain analysis. The relatively low correlation in WM is also likely to be partially driven by the mismatch between the tensor-based simplification of WM architecture as FA, which best represents a single fibre population, and the reality that most regions include multiple fibre populations [76]. It may be necessary to combine more fibre-specific parameters from multishell DWI [77] with fibre-specific T1/R1 estimates from inversion-recovery DWI [78,79] to more accurately specify this relationship. In addition, correlations with FA have been found to be stronger when restricted to anatomically defined tracts rather than across all white matter [34]. Our findings are in line with decades of work characterizing the relationships between MR metrics in WM and support the idea that differences/changes in FA may not generally be attributable to a specific anatomical or physiological property such as myelin [22,80].

The correlation between R1 and PD was stronger in GM than WM, and this effect was most prominent in subcortical GM, similarly to the R1-R2* relationship. We do not have a clear interpretation for this R1-PD relationship. However, previous literature suggests that PD and T1/R1 are related through their dependence on cell densities – which affects free water content [5]. Furthermore, the echo averaging prior to the map calculation introduces an R2* bias on the PD estimates [30], which may be a partial driver of the stronger correlation of R1 and PD in subcortical GM.

Our results when comparing WM with cortical and subcortical GM separately provide additional complementary evidence for the similarity and differences between tissue types. We found that the correlations between metrics in subcortical GM resembled more closely those found in WM than those found in cortical GM (mean t in cortical vs WM = 14.23, mean t in subcortical vs WM = 12.62). We also observed overall stronger pairwise correlations in GM (cortical and subcortical) when compared to WM, which could be due to many interacting factors. First, strong spatial cortical gradients (e.g., cytoarchitecture) may have driven some of the correlation strength in GM, as consistent changes in physiology could be translated to consistent and matched changes in multiple metrics. Additionally, the directional complexity of WM may lower the redundancy of certain metrics that are sensitive to directionality of fibers, especially in voxels with high heterogeneity. Lastly, many of the metrics used in this analysis have been specifically optimized to disentangle WM heterogeneity, such as FA, MD and R1. In fact, the lack of redundancy and therefore lower correlations may be explicitly due to the specificity of these metrics, as they are better able to measure physiological factors in WM than in GM. However, while this provides a holistic comparison of the metric-metric correlations across tissue types, correlations may vary across different regions within each tissue. For example, regional cortical differences in cyto- and myelo-architecture or tract-specific differences in myelination or density would be expected to alter the relationships that we have identified here. Future work could explore how metric-metric correlations spatially vary across tissues in healthy control populations as a normative comparison to identify more subtle region-specific changes that may serve as early biomarkers for disease. Complementary approaches (e.g., in white matter [81,82]) could extend this work to define spatially varying patterns of multivariate similarity in metric-metric correlations in specific tissues or regions in individual participants. Additionally, while the present work attempted to

capture the linear dependencies of these metrics to facilitate understanding and application of our results, an investigation into the non-linearity of some of the pairwise relationships could potentially identify more subtle relationships between physiological factors or metric redundancies. Finally, we emphasize that while all pairwise correlations were statistically significant, this should not be interpreted as an indication of biological importance. The high statistical power inherent to voxelwise analyses driven by large sample sizes means that even very weak correlations can reach significance. Therefore, we focus our interpretation on the strength and spatial distribution of correlation coefficients, which more meaningfully reflect shared biological variance across metrics.

In order to capitalize on these redundancies, methods such as MR Fingerprinting (MRF) and MR-STAT have seen increasing use in the field. These methods aim to jointly estimate multiple quantitative parameters (e.g., T1, T2, PD) from highly undersampled or non-traditional acquisition schemes by fitting entire signal evolutions to precomputed dictionaries or solving large-scale inverse problems, respectively [83,84]. Both these methods therefore leverage redundancies in MR signals, similarly to the analysis performed here. By quantifying and spatially characterizing the correlations found in this work, and identifying the principal components that capture shared variance, we provide normative constraints that could be used to inform the design, optimization, or interpretation of MRF and MR-STAT protocols, either by identifying minimally redundant parameter subsets or by characterizing expected normative covariance patterns. Though we did not implement MRF or MR-STAT directly, our findings support their foundational assumption: that multiple tissue-sensitive metrics contain overlapping information that can be jointly modeled.

We were also interested in exploring how dimensionality reduction could be used to determine whether combining metrics enhances details that are difficult or impossible to detect with single metrics [22]. Consistent with the pattern that we identified in the bivariate statistics, R1's high covariance with multiple metrics appeared in the PCA as a cluster of metrics defining PC1 (including high loadings on all metrics except R2* and accounting for 44% of the variance). This component could be a potential candidate for a more specific index of cortical myelination [4,62]. PC1, along with most other PCs, resembled gradients found in individual metrics when mapped onto the cortex, such as that of primary sensory and motor cortices seen in R1. However, the last PC, which accounted for ~3% of the variance in the data, provided a gradient containing some regions similar to those in the default-mode network (DMN) commonly identified in functional resting-state MRI [55,85]. Previous research has shown links between the DMN and structural connectivity exhibited by diffusion tensor imaging and tractography [86], and between functional connectivity and cortical qT1 as a proxy for intracortical myelin [44]. Following from this work, it would be interesting to explore if the gradients exhibited by the PCs are more directly related to functional connectivity and the DMN than any single modality. Whether or not our last PC corresponds to the DMN, we believe that it is interesting because it was the most dissimilar PC to individual metric gradients, suggesting that such multimodal tools can uncover subtle gradients that are overshadowed by other sources of variation in the data [22].

One major limitation of the present study is that it provides only a description of the relationship between qMRI metrics, but cannot comment on their fundamental relationship with physiological tissue parameters. Properly addressing this question will require more studies combining qMRI sequences with ground truth histological and biomarker data in the same brains/tissues [2,6], although even these approaches are limited due to postmortem factors that alter microstrucure. While our study used linear dimensionality reduction to allow for clear interpretation, it is likely that proper mapping from MRI to histological variables will require non-linear statistical methods as well. An approach similar to the recently proposed Morphometric Similarity Networks [35] could be applied to quantitative multiparametric imaging data to better specify structural covariance [87,88] and eventually link it to the underlying physiological parameters. In addition, future work with ultra-high resolution MRI could be used to investigate multiparametric differences in contrasts across cortical layers [45,54,65,89].

The current study describes the bivariate relationships between qMRI metrics and an initial exploration of how the data is grouped in multidimensional space. We found that there was a wide range of covariance between metrics that differed across tissue type, but that these relationships were stable across individuals. Our findings are a step towards

using multimodal metric combinations to better identify tissue characteristics and develop a more direct link between non-invasive MRI and physiology.

## Supporting information

**S1 Fig. Surface plots for the six metrics.** Median values from the cortical ribbon across all participants are plotted on the cortical surface. The range of values are indicated on the colorbar for each metric (pu = percentage units).
(TIF)

**S2 Fig. Distribution of $R^2$ values across participants for all pairwise correlations (GM in blue, WM in red).** This shows that the relationships between metrics were highly consistent across individuals.
(TIF)

**S3 Fig. Distribution of $R^2$ values across participants for all pairwise correlations (cortical GM in green, subcortical GM in indigo).** This shows that the relationships between metrics were highly consistent across individuals.
(TIF)

**S4 Fig. Distribution of $R^2$ values across participants for all pairwise correlations (cortical GM in green, WM in red).** This shows that the relationships between metrics were highly consistent across individuals.
(TIF)

**S5 Fig. Distribution of $R^2$ values across participants for all pairwise correlations (WM in red, subcortical GM in indigo).** This shows that the relationships between metrics were highly consistent across individuals.
(TIF)

**S6 Fig. Pairwise correlations in WM (above the diagonal) and subcortical GM (below the diagonal) for all participants' data.** P-values for all pairwise correlations were very small (<0.001), however were not stated explicitly as the small values of p do not necessarily relate to higher explained variance. $R^2$ values for each pairwise correlation are shown above their respective graphs Data along the diagonal shows the histograms for each metric in subcortical (purple) and WM (red). Metric thresholds were implemented for $R2^*$ and MD to reduce partial volume effects and artifacts, and are visible as sharp cutoffs in the figure (see Method). Pairwise density plots are depicted in log space to show the full range of data.
(TIF)

**S7 Fig. Pairwise correlations in WM (above the diagonal) and cortical GM (below the diagonal) for all participants' data.** P-values for all pairwise correlations were very small (<0.001), however were not stated explicitly as the small values of p do not necessarily relate to higher explained variance. $R^2$ values for each pairwise correlation are shown above their respective graphs Data along the diagonal shows the histograms for each metric in cortical (green) and WM (red). Metric thresholds were implemented for $R2^*$ and MD to reduce partial volume effects and artifacts, and are visible as sharp cutoffs in the figure (see Method). Pairwise density plots are depicted in log space to show the full range of data.
(TIF)

**S1 Table. Results of paired two-tailed Student's t-tests for all pairwise metric-metric correlation distributions.** Significant Bonferroni corrected p-values are bolded. The mean of the absolute value of the t-statistics and difference between the means of the r distributions for each pair of metric-metric correlations is also reported as an estimate of the magnitude of the difference between tissue types.
(DOCX)

## Author contributions

**Conceptualization:** Francis Carter, Alfred Anwander, Thomas Goucha, Angela D. Friederici, Antoine Lutti, Nikolaus Weiskopf, Christopher J. Steele.

**Data curation:** Francis Carter, Alfred Anwander, Thomas Goucha, Helyne Adamson, Antoine Lutti, Nikolaus Weiskopf.

**Formal analysis:** Francis Carter, Mathieu Johnson, Pierre-Louis Bazin, Christopher J. Steele.

**Funding acquisition:** Alfred Anwander, Angela D. Friederici, Nikolaus Weiskopf, Christopher J. Steele.

**Investigation:** Francis Carter, Thomas Goucha, Helyne Adamson, Claudine J. Gauthier, Christopher J. Steele.

**Methodology:** Francis Carter, Alfred Anwander, Thomas Goucha, Helyne Adamson, Angela D. Friederici, Antoine Lutti, Pierre-Louis Bazin, Christopher J. Steele.

**Project administration:** Alfred Anwander, Angela D. Friederici, Nikolaus Weiskopf, Christopher J. Steele.

**Resources:** Alfred Anwander, Angela D. Friederici, Pierre-Louis Bazin, Christopher J. Steele.

**Software:** Francis Carter, Pierre-Louis Bazin, Christopher J. Steele.

**Supervision:** Alfred Anwander, Claudine J. Gauthier, Nikolaus Weiskopf, Pierre-Louis Bazin, Christopher J. Steele.

**Validation:** Francis Carter, Christopher J. Steele.

**Visualization:** Francis Carter, Pierre-Louis Bazin.

**Writing – original draft:** Francis Carter, Christopher J. Steele.

**Writing – review & editing:** Francis Carter, Alfred Anwander, Mathieu Johnson, Thomas Goucha, Helyne Adamson, Angela D. Friederici, Antoine Lutti, Claudine J. Gauthier, Nikolaus Weiskopf, Pierre-Louis Bazin, Christopher J. Steele.

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
