## [Decision Letter · Decision Letter 0]

PONE-D-24-02122Assessing Quantitative MRI Techniques using Multimodal ComparisonsPLOS ONE

Dear Dr. Steele,

Thank you for submitting your manuscript to PLOS ONE. After careful consideration, we feel that it has merit but does not fully meet PLOS ONE’s publication criteria as it currently stands. Therefore, we invite you to submit a revised version of the manuscript that addresses the points raised during the review process.

The two reviewers suggested "major revision" and "reject", respectively. They raised a number of concerns that would be sufficient to reject the manuscript. I'm allowing the authors to decide if revising the manuscript for two reasons: 1) the topic is intersting and timely, and the dataset can be exploited to extract useful information, and 2) reviewer #2 suggested rejection also based on perceived lack of novelty. Note that PLOS One does not consider lack of novelty a stringent criterium. PLOS One considers indeed replication studies. However, the nature, rationale and aims of the study must be clearly described. As stated  by the official publication criteria, "If a submitted study replicates or is very similar to previous work, authors must provide a sound scientific rationale for the submitted work and clearly reference and discuss the existing literature. Submissions that replicate or are derivative of existing work will likely be rejected if authors do not provide adequate justification." This conditions must be strictly adhered to. In other words, authors must either show that the manuscript includes enough innovative content, or clearly state what are they replicating and why. Beyond this crucial issue, reviewers raised specific concerns, including incomplete/unclear analysis.Given that a revision would require significan effort to address all the concerns, authors (if deciding to revise) can contact me or the editorial office to arrange a new deadline. 

We look forward to receiving your revised manuscript.

Kind regards,

Federico Giove, PhD

Academic Editor

PLOS ONE

Journal Requirements:

2. Thank you for stating the following financial disclosure: "CJS was supported by the Natural Sciences and Engineering Research Council (NSERC: RGPIN-2020-06812, DGECR-2020-00146) and the Heart and Stroke Foundation of Canada New Investigator Award and Catalyst from the Canadian Institutes of Health Research (HNC 170723). 

PLB was supported by the NWO Vici grant (PI: Birte Forstmann). 

https://www.nwo.nl/en/researchprogrammes/nwo-talent-programme

CJG was supported by the Heart and Stroke Foundation of Canada New Investigator Award, Michal and Renata Hornstein Chair in Cardiovascular Imaging, and NSERC (DG: RGPIN-2015-04665).  

AA and AF received funding from the SPP2041 program "Computational Connectomics" of the German Research Foundation (DFG). 

NW has received funding from the European Research Council under the European Union's Seventh Framework Programme (FP7/2007-2013) / ERC grant agreement n° 616905; from the European Union's Horizon 2020 research and innovation programme under the grant agreement No 681094; from the BMBF (01EW1711A & B) in the framework of ERA-NET NEURON."  

3. Thank you for stating the following in the Acknowledgments Section of your manuscript: "CJS was supported by the Natural Sciences and Engineering Research Council (NSERC: RGPIN-2020-06812, DGECR-2020-00146) and the Heart and Stroke Foundation of Canada New Investigator Award and Catalyst from the Canadian Institutes of Health Research (HNC 170723). 

PLB was supported by the NWO Vici grant (PI: Birte Forstmann). 

https://www.nwo.nl/en/researchprogrammes/nwo-talent-programme

CJG was supported by the Heart and Stroke Foundation of Canada New Investigator Award, Michal and Renata Hornstein Chair in Cardiovascular Imaging, and NSERC (DG: RGPIN-2015-04665).  

AA and AF received funding from the SPP2041 program "Computational Connectomics" of the German Research Foundation (DFG). 

NW has received funding from the European Research Council under the European Union's Seventh Framework Programme (FP7/2007-2013) / ERC grant agreement n° 616905; from the European Union's Horizon 2020 research and innovation programme under the grant agreement No 681094; from the BMBF (01EW1711A & B) in the framework of ERA-NET NEURON."  

Please remove any funding-related text from the manuscript and let us know how you would like to update your Funding Statement. Currently, your Funding Statement reads as follows: "CJS was supported by the Natural Sciences and Engineering Research Council (NSERC: RGPIN-2020-06812, DGECR-2020-00146) and the Heart and Stroke Foundation of Canada New Investigator Award and Catalyst from the Canadian Institutes of Health Research (HNC 170723). 

PLB was supported by the NWO Vici grant (PI: Birte Forstmann). 

https://www.nwo.nl/en/researchprogrammes/nwo-talent-programme

CJG was supported by the Heart and Stroke Foundation of Canada New Investigator Award, Michal and Renata Hornstein Chair in Cardiovascular Imaging, and NSERC (DG: RGPIN-2015-04665).  

AA and AF received funding from the SPP2041 program "Computational Connectomics" of the German Research Foundation (DFG). 

NW has received funding from the European Research Council under the European Union's Seventh Framework Programme (FP7/2007-2013) / ERC grant agreement n° 616905; from the European Union's Horizon 2020 research and innovation programme under the grant agreement No 681094; from the BMBF (01EW1711A & B) in the framework of ERA-NET NEURON."  

"The Max Planck Institute for Human Cognitive and Brain Sciences has an institutional research agreement with Siemens Healthcare. NW holds a patent on acquisition of MRI data during spoiler gradients (US 10,401,453 B2). NW was a speaker at an event organized by Siemens Healthcare and was reimbursed for the travel expenses."

5. We note that you have indicated that there are restrictions to data sharing for this study. For studies involving human research participant data or other sensitive data, we encourage authors to share de-identified or anonymized data. However, when data cannot be publicly shared for ethical reasons, we allow authors to make their data sets available upon request. For information on unacceptable data access restrictions, please see http://journals.plos.org/plosone/s/data-availability#loc-unacceptable-data-access-restrictions. 

Reviewers' comments:

Reviewer's Responses to Questions

**Comments to the Author**

1. Is the manuscript technically sound, and do the data support the conclusions?

Reviewer #1: Yes

Reviewer #2: Yes

2. Has the statistical analysis been performed appropriately and rigorously? 

Reviewer #1: No

Reviewer #2: No

3. Have the authors made all data underlying the findings in their manuscript fully available?

Reviewer #1: No

Reviewer #2: No

4. Is the manuscript presented in an intelligible fashion and written in standard English?

Reviewer #1: Yes

Reviewer #2: Yes

5. Review Comments to the Author

Reviewer #1: The authors conducted quantitative multi-parametric and diffusion MRI in a reasonable sample size of 101 young healthy individuals (range 18-34 years). They report linear relationships between the metrics derived from these acquisitions in three tissue types (white matter, cortical grey matter, and subcortical grey matter). They then applied principal component analysis across the cortical voxels to derive latent variables and describe the cortical gradients as uncovered by the combinations of metrics loading onto the principal coordinates.

Overall, the manuscript is clearly written and is interesting. The results are interesting for researchers working with multi-modal imaging data and particularly qMRI.

General suggestions

1. The authors study metric-metric correlations in three tissues. Yet, the PCA was only conducted in the cortex. I suggest the authors replicate this to the other tissues (white matter and deep grey matter). This will be important given that some previous studies have investigated this in major white matter tracts, although with a different set of MRI parameters (e.g. combining diffusion data with ihMT and mcDESPOT by Geeraert et al 2020 https://doi.org/10.1371/journal.pone.0233244). It would be important to discuss if the resulting latent variables in white matter have similar interpretation to those in grey matter, or how similar are the latent variable interpretations in white matter derived using qMRI metrics compared to previous works. The authors state that the cortical PC6 was the most novel as it least resembled individual metrics – would all 6 PCs be informative in other tissues too? Is the more/less redundancies of the metrics in the different tissues? Do the components follow any known patterns of development or degeneration in the different tissues?

2. The diffusion data has potentially been under-utilized. There are several other models that can be derived from single shell data (e.g. https://www.sciencedirect.com/science/article/pii/S1053811922002956?via%3Dihub), which can add additional information. How do you expect the inclusion of more models to vary the PCA components? In addition, what is the rationale for excluding RD as it has shown some sensitivity to myelin-related pathologies (see also Blesa et al 2022 (https://doi.org/10.1162/imag_a_00017) demonstrating high correlations between RD and MTsat in white matter). In addition, do the authors think the acquisition of another shell may benefit the experiment?

Abstract

1. In the sentence starting with “To support the development ...” please add “and dMRI” metrics. This is because dMRI metrics can’t strictly be classified under qMRI.

Introduction

Overall enough background information is included in the Introduction section. However, some key references seem to be missing. Some specific comments and suggestions are as follows:

1. Please define all abbreviations at first use, e.g. WM, CSF, MTR.

2. The first sentence of the third paragraph of the Introduction needs to be clarified. The part before the brackets is confusing and contrasting with that included in the brackets. Perhaps paraphrase the entire sentence to: "One major issue is that the same MR signals can be the result of many different molecular arrangements and their concentrations"

3. In the third paragraph, when discussing how latent variables can be derived from MRI metrics, the authors should also reference Chamberland et al 2019 https://doi.org/10.1016/j.neuroimage.2019.06.020.

4. In the same sentence, the reference to Geeraert et al 2019 https://doi.org/10.1002/hbm.24706 seems misplaced as this paper did not derive latent variables, though they did analyse data from multiple acquisitions in relation to age and sex. Yet, in another paper by the same first author they did apply dimensionality reduction for dMRI, ihMT and mcDESPOT data and should be referenced instead: Geeraert et al 2020 https://doi.org/10.1371/journal.pone.0233244. Of note, the authors reference the 2019 paper in multiple places in the manuscript when referring to latent variables, so I suggest them to doublecheck the references.

5. The sentence starting with “qMRI has other advantages, including ...”: it should include “diffusion MRI commonly used for assessing microstructural integrity” as it’s not only structural MRI that qMRI provides advantages to.

6. In the 4th paragraph when discussing redundancies of information provided by MRI metrics, it is important to also cite Chamberland et al 2019, and De Santis et al 2014 https://doi.org/10.1016/j.neuroimage.2013.12.003

7. In the same paragraph, the sentence discussing previous work of metric-metric correlations could also include references to Chamberland et al 2019, Geeraert et al 2020, and Vaher et al 2022 https://doi.org/10.1016/j.neuroimage.2022.119169 - these papers have also explored metric-metric covariance within and along major white matter tracts (although two of these only focus on diffusion metrics, but exemplify the redundancy aspect). The authors could also add reference to Blesa et al 2022 (https://doi.org/10.1162/imag_a_00017) which investigated correlations between dMRI and MT-metrics in major WM tracts and across all WM voxels in a neonatal dataset.

Methods

Overall, the methods of this paper seem robust. To reduce partial volume effects, the authors apply two different tissue segmentation toolkits and limit their analyses for voxels included in the intersection of the methods. Other methods for cortical surface generation and group co-registration follow previously published methods. Some points for clarification:

1. Please specify which methods were used for susceptibility correction and image registration were used. Please also include references.

2. Section 2.2 last paragraph: the authors state that magnetisation transfer maps were estimated from MPMs. Please be more specific which exact MT metric was derived: was it magnetisation transfer ratio (MTR) or magnetisation transfer saturation (MTsat). Both seem to be possible to be derived from the data at hand. Arguably, MTsat would be a better measure given that it is inherently corrects for B1+ inhomogeneities and T1 relaxation to a substantial degree.

3. Section 2.3.1 segmentation: the authors use seemingly arbitrary thresholds for excluding voxels. Please provide references and/or reasoning for the use of these thresholds.

4. Section 2.5.1 statistical analysis: I think the regression models to evaluate metric-metric correlations should include adjustment for repeated measures within participant as each participant contributed to the analysis with values from thousands of voxels.

Results

1. Section 3.1 and Figure 1 legend should include a cross-reference to Supplementary figure 1.

2. Please include the figure with all PCA gradients in the main text (i.e. combine Fig 4 and supplementary fig 2) and include discussion about it in the Results and Discussion section.

Discussion

1. The last sentence of the first sentence should clarify that these results only apply to PCA conducted in the cortex.

2. Given that cortical and subcortical metric-metric correlations also differed, it is important to include a further discussion about all pairwise comparisons of the correlations between the three tissues. For example, are the correlations in WM more similar to those observed in subcortical GM? Or were cortical and subcortical GM correlations more similar to one another than compared to WM? Another angle to answer these questions would be to include the PCA in the three tissues, and compare and contrast the resulting radar plots.

3. How can these findings of MRI metric covariance be extrapolated to other species or developmental time points?

4. When discussing the correlations between FA and R1 it would be good to include a reference to Blesa et al 2022 using a neonatal dataset as it found something similar across all WM voxels (low correlations between FA and MTsat), while the correlations were different withing major tracts (FA correlations with MTsat stronger).

Reviewer #2: The authors report the correlation between standard qMRI parameters and selected DTI metrics (FA and MD) in grey matter and white matter regions. Data was collected on 101 young subjects with relatively small age variation hence minor age effects were introduced in the analysis. GM was divided into sub-cortical GM and cortical GM (voxels were extracted along a skeleton of cortical GM). Correlation results were as expected from the literature. The authors explored the usage of principal component analysis (PCA) of the parameters from qMRI and DTI and mapped the components back to the brain to see if they can explain anatomical relations. I believe it is a nice data set collected, and it is informative to inspect the correlations, but the scientific contribution is not clear to me. The discussion reads as a confirmation of the many existing studies and does not add a new dimension. The PCA seems interesting, but the authors did not take it further and it is a kind of hanging e.g. relating it to cortical regional (from atlas) and/or tract-based atlases allowing the drawing of some functional-related conclusions.

Section 2.5.2. please make clear what metrics refer to. The authors should elaborate on the PCA method part.

Figure 1: Please add arrows for the reader to follow the explanation, especially the cortex part. The figure text also needs to be elaborated.

P18. “… showing little to no partial voluming …” in relation to what?

Section 3.4. The authors refer several times in the text to a “gradient” but it is unclear what this refers to.

P18 “…excellent …” What do the authors mean by “excellent”?

Section 3.2 and figure 2. The authors state that all metrics are significantly correlated but when inspecting the correlation plot in Figure 2 is not convincingly supporting the result. E.g. MT vs FA and PD vs PD etc. I suggest showing the correlation in each plot of Figures 2 and 3 to support the statistics.

P21 ” …such as Alzheimer’s tau pathologies and white matter hyperintensities,…” Please elaborate.

P22: “Interestingly, the R1-R2* correlation was weaker in WM where we might expect…” Please elaborate on this sentence. It is unclear to me why this is the case.

6. PLOS authors have the option to publish the peer review history of their article (what does this mean? ). If published, this will include your full peer review and any attached files.

**Do you want your identity to be public for this peer review?** For information about this choice, including consent withdrawal, please see our Privacy Policy .

Reviewer #1: **Yes: ** Kadi Vaher

Reviewer #2: No

---

## [Author Response · Author response to Decision Letter 1]

20 Nov 2024

NOTE: the response to reviewers has also been provided in a formatted word document

Reviewer #1: The authors conducted quantitative multi-parametric and diffusion MRI in a reasonable sample size of 101 young healthy individuals (range 18-34 years). They report linear relationships between the metrics derived from these acquisitions in three tissue types (white matter, cortical grey matter, and subcortical grey matter). They then applied principal component analysis across the cortical voxels to derive latent variables and describe the cortical gradients as uncovered by the combinations of metrics loading onto the principal coordinates.

Overall, the manuscript is clearly written and is interesting. The results are interesting for researchers working with multi-modal imaging data and particularly qMRI.

General suggestions

1. The authors study metric-metric correlations in three tissues. Yet, the PCA was only conducted in the cortex. I suggest the authors replicate this to the other tissues (white matter and deep grey matter). This will be important given that some previous studies have investigated this in major white matter tracts, although with a different set of MRI parameters (e.g. combining diffusion data with ihMT and mcDESPOT by Geeraert et al 2020 https://doi.org/10.1371/journal.pone.0233244). It would be important to discuss if the resulting latent variables in white matter have similar interpretation to those in grey matter, or how similar are the latent variable interpretations in white matter derived using qMRI metrics compared to previous works. The authors state that the cortical PC6 was the most novel as it least resembled individual metrics – would all 6 PCs be informative in other tissues too? Is the more/less redundancies of the metrics in the different tissues? Do the components follow any known patterns of development or degeneration in the different tissues?

• Thank you for this excellent comment that contains some of the rationale for additional exciting approaches that we are currently working on. We are also very interested in white matter microstructure and recently released a paper that tackles multivariate white matter in a more specific way than we were able to do in the current manuscript. This work (Tremblay et al., 2024) used a specific white matter registration approach (FOD-based registrations) to ensure that white matter alignment was optimized across individuals. We had a more comprehensive set of WM metrics (DTI, NODDI, CSD, T1w/T2w) and feel that this is a much stronger basis for comparison within white matter. As a result, our group is currently exploring analyses similar to those that the reviewer has described here in the dataset described in the Tremblay paper. Similar to the paper of Geeraert et al. (2020), we are also looking to see how latent components are related to behavioural differences across a large number of individuals. A robust investigation of the important questions that the reviewer has raised here is unfortunately beyond the scope of the current manuscript. However, we also agree that they are extremely important and are in the process of addressing them with a more comprehensive dataset combining additional microstructural measurements and comprehensively characterized multi-domain behavioural data in a large sample of individuals (>700). Additionally, the main focus for the current work was to provide a comprehensive description of the metrics’ linear relationships and, as a result, in the current manuscript we have therefore chosen not to expand on the PCA analysis in the WM and subcortical GM. However, based on this comment (and others) we have now included the full comparisons between WM, cortical GM, and subcortical GM and included an additional analysis to provide an initial assessment for how similar the microsctructural profiles in each tissue type are. Please refer to our reply to Point 2 in the Discussion section of the current reviewer’s comments below for the details about these additions.

• To provide additional context on our thinking and as a result of the reviewers’ interest in this topic, we include here a short description of our approach along with some exemplary references. We proposed the Mahalanobis distance as an approach to characterize microstructural differences between individuals and/or groups while accounting for the intrinsic covariance between metrics. This approach provides a single value at each voxel (the Mahalanobis distance) that represents the overall multivariate microstructural deviation from a reference. It allows the inclusion of as many metrics as are available to the researcher without penalty and has shown excellent promise in identifying microstructural differences between groups (see example references included below). In cases where we (and other researchers) are not specifically interested in the metric loadings (i.e., the PC loadings for each individual) the Mahalanobis distance is a simple way to identify microstructural deviation.

Tremblay SA, Alasmar Z, Pirhadi A, Carbonell F, Iturria-Medina Y, Gauthier CJ, Steele CJ. 2024. MVComp toolbox: MultiVariate Comparisons of brain MRI features accounting for common information across measures. Aperture Neuro. 4.

Owen TW, de Tisi J, Vos SB, Winston GP, Duncan JS, Wang Y, et al. Multivariate white matter alterations are associated with epilepsy duration. European Journal of Neuroscience 2021;53:2788–803. https://doi.org/10.1111/ejn.15055.

Gyebnár G, Klimaj Z, Entz L, Fabó D, Rudas G, Barsi P, et al. Personalized microstructural evaluation using a Mahalanobis-distance based outlier detection strategy on epilepsy patients’ DTI data – Theory, simulations and example cases. PLOS ONE 2019;14:e0222720. https://doi.org/10.1371/journal.pone.0222720.

Guerrero-Gonzalez JM, Yeske B, Kirk GR, Bell MJ, Ferrazzano PA, Alexander AL. Mahalanobis distance tractometry (MaD-Tract) – a framework for personalized white matter anomaly detection applied to TBI. NeuroImage 2022;260:119475. https://doi.org/10.1016/j.neuroimage.2022.119475.

2. The diffusion data has potentially been under-utilized. There are several other models that can be derived from single shell data (e.g. https://www.sciencedirect.com/science/article/pii/S1053811922002956?via%3Dihub), which can add additional information. How do you expect the inclusion of more models to vary the PCA components? In addition, what is the rationale for excluding RD as it has shown some sensitivity to myelin-related pathologies (see also Blesa et al 2022 (https://doi.org/10.1162/imag_a_00017) demonstrating high correlations between RD and MTsat in white matter). In addition, do the authors think the acquisition of another shell may benefit the experiment?

• This point is well taken. While we agree that there are additional WM metrics that could be included, we also recognize that their inclusion would significantly increase the number of multiple comparisons calculated. To avoid p-hacking, we chose our dMRI metrics a priori from the diffusion tensor (FA and MD) to represent the most commonly used metrics in human MRI studies that utilize the diffusion tensor to analyze dMRI. While not exhaustive, our results with FA and MD will be informative for other researchers using, or planning to use, qMRI in their work in both GM and WM.

• For whether or not the PCA results would change with additional metrics, we expect that adding additional metrics may shift relationships somewhat, but that the general pattern would remain similar when the approach is used in healthy adults. Beyond questions in the healthy adult population, this is perhaps more interesting to consider in the context of disease and/or development – as we may expect a shift in the correlations between metrics as a result of a differential effect of pathology/development on specific biophysical tissue parameters (e.g., myelin, cell density). In this case, it may be more beneficial to include as many metrics as possible to maximize the chance that one of them will reflect pathophysiological (or developmental) change and therefore result in a differential weighting on the PCs. With this type of data, we would therefore hypothesize that adding an additional shell to the dMRI acquisition would allow additional metrics to be calculated, that may be more sensitive to the expected change. As detailed in our previous response above, while exploring all of these questions is beyond the scope of the current manuscript, we have begun to tackle them with a complementary approach in some of our other recent work (Tremblay et al., 2024).

Abstract

1. In the sentence starting with “To support the development ...” please add “and dMRI” metrics. This is because dMRI metrics can’t strictly be classified under qMRI.

• This has been clarified in the text.

◦ To support the development and common use of qMRI for cognitive neuroscience, we analysed a set of qMRI and dMRI metrics (R1, R2*, Magnetization Transfer saturation, Proton Density saturation, Fractional Anisotropy, Mean Diffusivity) in 101 healthy young adults.

Introduction

Overall enough background information is included in the Introduction section. However, some key references seem to be missing. Some specific comments and suggestions are as follows:

1. Please define all abbreviations at first use, e.g. WM, CSF, MTR.'

• Thank you for pointing this out. We have now defined all abbreviations at first use.

2. The first sentence of the third paragraph of the Introduction needs to be clarified. The part before the brackets is confusing and contrasting with that included in the brackets. Perhaps paraphrase the entire sentence to: "One major issue is that the same MR signals can be the result of many different molecular arrangements and their concentrations"

• This sentence has been clarified. We have changed the wording as follows:

◦ One major issue is that the MR signal from a single voxel is potentially the result of many different molecular arrangements and concentrations, and therefore provides ambiguous information about the microstructural composition in that voxel [Draganski et al., 2011; Tardif et al., 2016; Tardif et al., 2017].

3. In the third paragraph, when discussing how latent variables can be derived from MRI metrics, the authors should also reference Chamberland et al 2019 https://doi.org/10.1016/j.neuroimage.2019.06.020.

• Thank you for this suggestion. We have integrated this into the text.

◦ Extracted latent variables can be related to microstructural features and ground-truth molecular concentrations to determine how well they map to specific tissue properties, such as disentangling microstructural differences related to diffusion hinderance from those related to orientation and fiber organization [Borsboom et al., 2003; (Chamberland et al., 2019); Filo et al., 2019; (Geeraert et al., 2020)].

4. In the same sentence, the reference to Geeraert et al 2019 https://doi.org/10.1002/hbm.24706 seems misplaced as this paper did not derive latent variables, though they did analyse data from multiple acquisitions in relation to age and sex. Yet, in another paper by the same first author they did apply dimensionality reduction for dMRI, ihMT and mcDESPOT data and should be referenced instead: Geeraert et al 2020 https://doi.org/10.1371/journal.pone.0233244. Of note, the authors reference the 2019 paper in multiple places in the manuscript when referring to latent variables, so I suggest them to doublecheck the references.

• Thank you for catching this referencing error. We have corrected the in-text citations and bibliography to reflect this.

5. The sentence starting with “qMRI has other advantages, including ...”: it should include “diffusion MRI commonly used for assessing microstructural integrity” as it’s not only structural MRI that qMRI provides advantages to.

• Thank you for pointing this out, this has now been clarified in the text as follows:

◦ qMRI has other advantages, including higher reproducibility and comparability between acquisitions within and between participants than conventional structural MRI commonly used for assessing macroscopic morphological change, as well as diffusion MRI commonly used for assessing microstructural integrity [Caan et al., 2019; Leutritz et al., 2020; Weiskopf et al., 2015].

6. In the 4th paragraph when discussing redundancies of information provided by MRI metrics, it is important to also cite Chamberland et al 2019, and De Santis et al 2014 https://doi.org/10.1016/j.neuroimage.2013.12.003

• We have responded to this point in our reply to point 7 below.

7. In the same paragraph, the sentence discussing previous work of metric-metric correlations could also include references to Chamberland et al 2019, Geeraert et al 2020, and Vaher et al 2022 https://doi.org/10.1016/j.neuroimage.2022.119169 - these papers have also explored metric-metric covariance within and along major white matter tracts (although two of these only focus on diffusion metrics, but exemplify the redundancy aspect). The authors could also add reference to Blesa et al 2022 (https://doi.org/10.1162/imag_a_00017) which investigated correlations between dMRI and MT-metrics in major WM tracts and across all WM voxels in a neonatal dataset.

• We have adapted this paragraph to include these references as well as the references mentioned in point 6. Please find the paragraph in its entirety below for direct reference. Just to note that the Blesa reference is cited by the author’s last name below (Cábez).

◦ While useful for acquiring multiple co-registered modalities, and often more robust in explaining variation in brain microstructure [De Santis et al., 2014], multiparametric methods likely contain redundancies in the information provided by each metric [Callaghan et al., 2015b; Weiskopf et al., 2015; Weiskopf et al., 2021], especially given the fact that dMRI and qMRI metrics show high correlation with eachother [Cábez et al., 2023]. In fact, in certain circumstances, a subset of qMRI and dMRI metrics that show redundancies to one other metrics present in the analysis can be removed without significantly reducing the variance explained by the remaining metrics. As such, these redundancies have the potential to be exploited to determine which combination of metrics accounts for the most variance in the tissue of interest (and could therefore form a basis for a minimum useful multiparametric acquisition profile) and, importantly, can then be used to map putative microstructural similarities across the brain [Weiskopf et al., 2015]. Similar to the way that microstructure exhibits differences across different tissues and regions, the relationships between metrics are also expected to vary. Previous work has explored metric-metric covariance relationships with ROI-based or segmented network approaches [Seidlitz et al., 2018; Uddin et al., 2019] such as analyzing metric covariance in predetermined WM tracts of interest [Chamberland et al., 2019; Geeraert et al., 2020; Vaher et al., 2022], and a within-ROI binning approach [Filo et al., 2019], which rely on a priori regional delineations and averaging that may obscure subtle variability within regions. A voxel-wise approach could provide a more nuanced and comprehensive description of these relationships and help to determine if and/or how multiparametric combinations can provide additional information not found in individual metrics [Paquola et al., 2019].

Methods

Overall, the methods of this paper seem robust. To reduce partial volume effects, the authors apply two different tissue segmentation toolkits and limit their analyses for voxels included in the intersection of the methods. Other methods for cortical surface generation and group co-registration follow previously published methods. Some points for clarification:

1. Please specify which methods were used for susceptibility correction and image registration were used. Please also include references.

• Thank you for pointing out this omission. The susceptibility correction and the image registration were both conducted using tools in the FSL toolkit. Specifically, the registration was conducted using F

---

## [Decision Letter · Decision Letter 1]

PONE-D-24-02122R1Assessing Quantitative MRI Techniques using Multimodal ComparisonsPLOS ONE

Dear Dr. Steele,

Thank you for submitting your manuscript to PLOS ONE. After careful consideration, we feel that it has merit but does not fully meet PLOS ONE’s publication criteria as it currently stands. Therefore, we invite you to submit a revised version of the manuscript that addresses the points raised during the review process.

We look forward to receiving your revised manuscript.

Kind regards,

Md Nasir Uddin, PhD

Academic Editor

PLOS ONE

Journal Requirements:

Reviewers' comments:

Reviewer's Responses to Questions

**Comments to the Author**

1. If the authors have adequately addressed your comments raised in a previous round of review and you feel that this manuscript is now acceptable for publication, you may indicate that here to bypass the “Comments to the Author” section, enter your conflict of interest statement in the “Confidential to Editor” section, and submit your "Accept" recommendation.

Reviewer #3: (No Response)

Reviewer #4: (No Response)

2. Is the manuscript technically sound, and do the data support the conclusions?

Reviewer #3: Yes

Reviewer #4: Partly

3. Has the statistical analysis been performed appropriately and rigorously? 

Reviewer #3: Yes

Reviewer #4: Yes

4. Have the authors made all data underlying the findings in their manuscript fully available?

Reviewer #3: No

Reviewer #4: Yes

5. Is the manuscript presented in an intelligible fashion and written in standard English?

Reviewer #3: Yes

Reviewer #4: Yes

6. Review Comments to the Author

Reviewer #3: This study examines quantitative multi-parametric and diffusion MRI, categorizing the linear relationships between derived imaging metrics across three tissue types. Additionally, the authors performed principal component analysis on cortical voxels, identifying latent variables that represent cortical gradients.

Overall, this work is coherent and cohesive in the presentation and analysis. Important clarifications to the text and figures have been made already through a first round of revisions. Because of this I have no major revisions to suggest, just some minor comments.

General Comments:

• A previous reviewer noted that ‘MT’ should be specified as ‘MT Saturation’. While this has been implemented, there is inconsistency in the use of ‘MTSat’ and ‘MT’ throughout the manuscript. Please standardise this.

• Readability of Figures 2, S6, S7 could be improved by increasing the font size of the axes and statistics.

Methods:

The authors account for multiple factors, suggesting an interest in controlling for potential plasticity influences. However, it is unclear why parameters such as participant sex and weight were not included as control variables. A brief justification for this omission would be helpful.

Papers of potential interest:

M. Ingalhalikar, A. Smith, D. Parker, T.D. Satterthwaite, M.A. Elliott, K. Ruparel, H. Hakonarson, R.E. Gur, R.C. Gur, & R. Verma, Sex differences in the structural connectome of the human brain, Proc. Natl. Acad. Sci. U.S.A. 111 (2) 823-828, https://doi.org/10.1073/pnas.1316909110 (2014).

Tian, L., Wang, L., Li, Q., Yan, C.-G., Ding, J.-H., & Wang, Y.-F. (2020). Gender differences are encoded differently in the structure and function of the human brain revealed by multimodal MRI. Frontiers in Human Neuroscience, 14, 244. https://doi.org/10.3389/fnhum.2020.00244

Dietze, L. M. F., McWhinney, S. R., Radua, J., & Hajek, T. (2023). Extended and replicated white matter changes in obesity: Voxel-based and region of interest meta-analyses of diffusion tensor imaging studies. Frontiers in Nutrition, 10, 1108360. https://doi.org/10.3389/fnut.2023.1108360

Kullmann, S., Schweizer, F., Veit, R., Fritsche, A., & Preissl, H. (2015). Compromised white matter integrity in obesity. Obesity reviews : an official journal of the International Association for the Study of Obesity, 16(4), 273–281. https://doi.org/10.1111/obr.12248

• In the sentence 'masks were inspected for each individual by FC,' it is unclear upon initial reading whether 'FC' refers to an author or a software tool. Please clarify by rewording or using initials with periods (e.g., F.C.).

Discussion:

• A brief discussion on other multi-parametric MRI methods that leverage these redundancies, such as MR Fingerprinting or MR-STAT, would strengthen the manuscript. While MRF is referenced once, its relevance to the current study is not fully elaborated.

Reviewer #4: Review: This study sought to investigate the multi-modal relationships between several quantitative MRI metrics, including R1, R2*, MT, PD, FA, MD). Additionally, the authors perform a PCA on the metrics in order to reduce dimensionality and examine whether combinations of metrics may be informative. Interestingly, the authors observed higher pair-wise correlations in areas of gray matter than white matter and also suggest that lower variance components of the PCA may provide unique information about the brain. Overall, the paper is well written, and I think the authors do a good job addressing previous reviews. However, I do have some additional comments:

Comments:

1) In the abstract, the statement that qMRI “provides normative values for comparisons between tissues, regions, and individuals” – while I agree, I’m not sure “normative” is correct as this would suggest comparison within a normative sample. However, if examining clinical features or group differences, then values wouldn’t be considered normative.

2) Minor typo: Pg. 3 of introduction: “especially given the fact that dMRI and qMRI metrics show high correlation with eachother” – each other.

3) It is interesting that all the pairs of correlations were significant given the large range of R2 values. For example, I would not have thought that the MD-R2* correlation (R2=0.001, R=0.032) would be significant. When computing these correlations, are average values across WM/GM being used or are all voxels within WM/GM being used for all participants? If the latter, could the significance arise more from the large “sample” being used to compute the correlations?

4) It is surprising that the authors observe higher pairwise correlations within GM compared to WM. Why do the authors think this is? Given the focus of DTI and relaxometry methods traditionally in WM, I would have thought that these pairwise correlations would have been higher in WM.

5) In the discussion, the authors state “quantitative metrics have wide variations in their degree of correlation with each other, but are consistent across participants.” What is meant by “consistent across participants”?

6) The authors mention in several places about the sensitivity of the qMRI metrics to age. Given there is a broad age range in the sample, does age influence the relationships observed here?

7) Did the authors test whether the relationships between metrics were non-linear?

7. PLOS authors have the option to publish the peer review history of their article (what does this mean? ). If published, this will include your full peer review and any attached files.

**Do you want your identity to be public for this peer review?** For information about this choice, including consent withdrawal, please see our Privacy Policy .

Reviewer #3: **Yes: ** Emma Louise Thomson

Reviewer #4: No

---

## [Author Response · Author response to Decision Letter 2]

24 Apr 2025

Reviewer Comments

Reviewer #3: This study examines quantitative multi-parametric and diffusion MRI, categorizing the linear relationships between derived imaging metrics across three tissue types. Additionally, the authors performed principal component analysis on cortical voxels, identifying latent variables that represent cortical gradients.

Overall, this work is coherent and cohesive in the presentation and analysis. Important clarifications to the text and figures have been made already through a first round of revisions. Because of this I have no major revisions to suggest, just some minor comments.

General Comments:

• A previous reviewer noted that ‘MT’ should be specified as ‘MT Saturation’. While this has been implemented, there is inconsistency in the use of ‘MTSat’ and ‘MT’ throughout the manuscript. Please standardise this.

• Thank you for pointing out this inconsistency. We have corrected all instances of MT to MTSat, where applicable. In some cases, where we refer to Magnetization Transfer in general, we have left the acronym as MT, but clarified this in the text.

Readability of Figures 2, S6, S7 could be improved by increasing the font size of the axes and statistics.

• The font sizes have been increased and the subplots have been spaced out to increase clarity.

Methods:

The authors account for multiple factors, suggesting an interest in controlling for potential plasticity influences. However, it is unclear why parameters such as participant sex and weight were not included as control variables. A brief justification for this omission would be helpful.

• We thank the reviewer for this suggestion. We understand that the omission of age (as raised by Reviewer 4 #6), sex, and weight as covariates was not sufficiently explained, and we have now addressed this with a short justification in the methods. Unfortunately, weight data was not collected for this dataset. There are (as the reviewer has pointed out) many instances where sex is an important variable to consider when assessing microstructural differences and/or relationships with behavioural variables or patient outcomes. However, our analytical goal with this research was not to characterize individual differences or model group effects, but rather to characterize normative patterns in multimodal MRI signals. As such, we attempted to ensure that the data used in the analysis was of high quality and consistency, but that controlling for individual variability was not prioritized. In other words, we did not assess individual quantitative metrics, but rather identified the linear pair-wise relationships between them. From this bivariate perspective, we have no reason to believe that the relationships between the physiological factors that each metric is sensitive to would be different depending on individual characteristics. As these quantitative markers sample from a subset of physiological factors, we expect that any differences are distributed across metrics and therefore do not alter their linear relationships. However, with a significantly larger and sex-balanced dataset and/or age range it may be possible to tease out subtle differences (for example, as a way to probe for normative vs. pathological age-related changes) and could be an interesting future direction. Lastly, we observed strikingly low variance of correlation strengths across participants, as discussed further below, which supports the hypothesis that accounting for age and sex would not significantly impact our results, as there was very little change in metric-metric relationships from participant to participant. As suggested by the reviewer, we have added additional text to the method to explain our rationale:

◦ “While age and sex differences have been shown to have an impact on brain structure (Ingalhalikar et al., 2014; Tian et al., 2020), this analysis did not include these variables as covariates. In our analyses we aimed to identify the linear pair-wise relationships between metrics, which should be relatively robust to these differences, rather than assess individual quantitative metrics. Furthermore, many of the metrics used here sample from a subset of physiological markers, hence their redundancies. This suggests that the differences due to age and sex would be distributed across the metrics of interest, which would limit their impact on the pairwise correlations we computed. In future research with a larger, sex-balanced, sample that spans a larger age range, subtle differences in these physiological factors could be explored”

Papers of potential interest:

M. Ingalhalikar, A. Smith, D. Parker, T.D. Satterthwaite, M.A. Elliott, K. Ruparel, H. Hakonarson, R.E. Gur, R.C. Gur, & R. Verma, Sex differences in the structural connectome of the human brain, Proc. Natl. Acad. Sci. U.S.A. 111 (2) 823-828, https://doi.org/10.1073/pnas.1316909110 (2014).

Tian, L., Wang, L., Li, Q., Yan, C.-G., Ding, J.-H., & Wang, Y.-F. (2020). Gender differences are encoded differently in the structure and function of the human brain revealed by multimodal MRI. Frontiers in Human Neuroscience, 14, 244. https://doi.org/10.3389/fnhum.2020.00244

Dietze, L. M. F., McWhinney, S. R., Radua, J., & Hajek, T. (2023). Extended and replicated white matter changes in obesity: Voxel-based and region of interest meta-analyses of diffusion tensor imaging studies. Frontiers in Nutrition, 10, 1108360. https://doi.org/10.3389/fnut.2023.1108360

Kullmann, S., Schweizer, F., Veit, R., Fritsche, A., & Preissl, H. (2015). Compromised white matter integrity in obesity. Obesity reviews : an official journal of the International Association for the Study of Obesity, 16(4), 273–281. https://doi.org/10.1111/obr.12248

• In the sentence 'masks were inspected for each individual by FC,' it is unclear upon initial reading whether 'FC' refers to an author or a software tool. Please clarify by rewording or using initials with periods (e.g., F.C.).

• Thank you for catching this. We have clarified that this refers to the initials of the first author (F.C., Francis Carter).

Discussion:

• A brief discussion on other multi-parametric MRI methods that leverage these redundancies, such as MR Fingerprinting or MR-STAT, would strengthen the manuscript. While MRF is referenced once, its relevance to the current study is not fully elaborated.

• We thank the reviewer for this excellent suggestion. We agree that further contextualization of our findings within the established methods helps to strengthen the manuscript and have added an additional paragraph in the discussion to situate our research in the context of these other multi-parametric methods.

• In order to capitalize on these redundancies, methods such as MR Fingerprinting (MRF) and MR-STAT have seen increasing use in the field. These methods aim to jointly estimate multiple quantitative parameters (e.g., T1, T2, PD) from highly undersampled or non-traditional acquisition schemes by fitting entire signal evolutions to precomputed dictionaries or solving large-scale inverse problems, respectively (Ma et al., 2013; Sbrizzi et al., 2018). Both these methods therefore leverage redundancies in MR signals, similarly to the analysis performed here. By quantifying and spatially characterizing the correlations found in this work, and identifying the principal components that capture shared variance, we provide normative constraints that could be used to inform the design, optimization, or interpretation of MRF and MR-STAT protocols, either by identifying minimally redundant parameter subsets or by characterizing expected normative covariance patterns. Though we did not implement MRF or MR-STAT directly, our findings support their foundational assumption: that multiple tissue-sensitive metrics contain overlapping information that can be jointly modeled.

Reviewer #4: Review: This study sought to investigate the multi-modal relationships between several quantitative MRI metrics, including R1, R2*, MT, PD, FA, MD). Additionally, the authors perform a PCA on the metrics in order to reduce dimensionality and examine whether combinations of metrics may be informative. Interestingly, the authors observed higher pair-wise correlations in areas of gray matter than white matter and also suggest that lower variance components of the PCA may provide unique information about the brain. Overall, the paper is well written, and I think the authors do a good job addressing previous reviews. However, I do have some additional comments:

Comments:

1) In the abstract, the statement that qMRI “provides normative values for comparisons between tissues, regions, and individuals” – while I agree, I’m not sure “normative” is correct as this would suggest comparison within a normative sample. However, if examining clinical features or group differences, then values wouldn’t be considered normative.

• Thank you for this comment. We agree that the use of the word “normative” here may lead to confusion. We have clarified as follows:

• “provides directly comparable values between tissues, regions, and individuals

2) Minor typo: Pg. 3 of introduction: “especially given the fact that dMRI and qMRI metrics show high correlation with eachother” – each other.

• Thank you for pointing out this typo. We have corrected it in the manuscript.

3) It is interesting that all the pairs of correlations were significant given the large range of R2 values. For example, I would not have thought that the MD-R2* correlation (R2=0.001, R=0.032) would be significant. When computing these correlations, are average values across WM/GM being used or are all voxels within WM/GM being used for all participants? If the latter, could the significance arise more from the large “sample” being used to compute the correlations?

• As the reviewer correctly notes, the significance of even extremely low correlation coefficients (with some of the R² values > 0.001) can be attributed in part to the very large number of voxels included in the analysis. To clarify, all voxelwise correlations were computed across all voxels within each tissue class (WM or GM), aggregated across all participants. As such, the very large sample size substantially increases statistical power, making even minimal associations statistically significant. We fully agree that statistical significance in this context should not be conflated with practical or biological significance. For this reason, we explicitly chose not to emphasize p-values in our main text or visualizations, and instead focused our interpretation on effect size, namely, the strength and distribution of correlation coefficients across metric pairs. This emphasis is intended to highlight the relative differences in inter-metric covariance, which was central to our goal of assessing metric redundancy and dimensional structure. To avoid potential confusion, we have slightly revised the figure caption and discussion text to reinforce this distinction between statistical and practical significance, and we thank the reviewer again for prompting this clarification.

• Finally, we emphasize that while all pairwise correlations were statistically significant, this should not be interpreted as an indication of biological importance. The high statistical power inherent to voxelwise analyses driven by large sample sizes means that even very weak correlations can reach significance. Therefore, we focus our interpretation on the strength and spatial distribution of correlation coefficients, which more meaningfully reflect shared biological variance across metrics.

4) It is surprising that the authors observe higher pairwise correlations within GM compared to WM. Why do the authors think this is? Given the focus of DTI and relaxometry methods traditionally in WM, I would have thought that these pairwise correlations would have been higher in WM.

• We thank the reviewer for raising this important point. We also initially thought that it was surprising that pairwise correlations between MRI metrics were consistently higher in gray matter (GM) than in white matter (WM), particularly given that many of these metrics, especially those derived from diffusion tensor imaging and relaxometry, were originally developed with WM microstructure in mind. We believe this pattern likely reflects several interacting factors. The presence of strong spatial cortical gradients (e.g., cytoarchitecture) may be driving some of the correlation strength in GM, as spatially consistent transitions in tissue properties may be artificially increasing metric covariance. In contrast, WM metrics are more spatially localized and sensitive to fiber orientation, which may serve to reduce global covariance. Additionally, as the reviewer notes, WM possesses high directional coherence, and many MRI metrics, particularly those derived from diffusion MRI, are sensitive to orientation dispersion, crossing fibers, and subvoxel heterogeneity. This may reduce correlations between metrics in WM, particularly when local tract complexity is high, even if overall tissue composition is relatively uniform. It is also the case that metrics optimized for WM microstructure (e.g., FA, MD, R1) are better tuned to isolate distinct biophysical properties in WM, and therefore may be more redundant in GM. In fact, a lack of redundancy between metrics in the WM would be a good indicator of sequences/metrics that were designed to more specifically target individual physiological parameters (in WM). This specificity may result in lower shared variance and thus lower inter-metric correlations. In GM, where these metrics were not initially designed to disentangle overlapping features, redundancy may be higher. We agree that this is an interesting and important point to discuss, and have now added text to present this within the discussion:

• “We also observed overall stronger pairwise correlations in GM (cortical and subcortical) when compared to WM, which could be due to many interacting factors. First, strong spatial cortical gradients (e.g., cytoarchitecture) may have driven some of the correlation strength in GM, as consistent changes in physiology could be translated to consistent and matched changes in multiple metrics. Additionally, the directional complexity of WM may lower the redundancy of certain metrics that are sensitive to directionality of fibers, especially in voxels with high heterogeneity. Lastly, many of the metrics used in this analysis have been specifically optimized to disentangle WM heterogeneity, such as FA, MD and R1. In fact, the lack of redundancy and therefore lower correlations may be explicitly due to the specificity of these metrics, as they are better able to measure physiological factors in WM than in GM.”

5) In the discussion, the authors state “quantitative metrics have wide variations in their degree of correlation with each other, but are consistent across participants.” What is meant by “consistent across participants”?

• This refers to the observation that, despite the variability in correlation strength between different pairs of metrics (Figures 2, 3), the pattern of cross-participant correlations for each metric is highly consistent (Supplementary Figures 3, 4,5). That is, when performing voxelwise analyses across all participants, each pair of metrics (and in each tissue type) had different correlations values (ranging from 0.0037 to 0.4465, Figures 2, 3) while the within-participant correlation values were highly consistent across subjects, as supported by the sharp distributions (i.e., low standard deviations) of the R2 values in Supplementary Figures 3, 4 and 5 We believe this distinction is important: it suggests that while the strength of the pairwise relationships varies widely between different pairs of metrics, the relationships of these pairs is consistent across participants in the sample. To address this, we have revised the manuscript text to clarify that “consistency across participants” refers to the reproducibility of metric-metric correlations across individuals.

• “Bivariate correlations between metrics were performed for each tissue type, and our results showed that quantitative metrics have wide variations in their degree of correlat

---

## [Decision Letter · Decision Letter 2]

Assessing Quantitative MRI Techniques using Multimodal Comparisons

PONE-D-24-02122R2

Dear Dr. Steele,

We’re pleased to inform you that your manuscript has been judged scientifically suitable for publication and will be formally accepted for publication once it meets all outstanding technical requirements.

Kind regards,

Md Nasir Uddin, PhD

Academic Editor

PLOS ONE

Additional Editor Comments (optional):

Reviewers' comments:

Reviewer's Responses to Questions

**Comments to the Author**

1. If the authors have adequately addressed your comments raised in a previous round of review and you feel that this manuscript is now acceptable for publication, you may indicate that here to bypass the “Comments to the Author” section, enter your conflict of interest statement in the “Confidential to Editor” section, and submit your "Accept" recommendation.

Reviewer #3: All comments have been addressed

Reviewer #5: All comments have been addressed

2. Is the manuscript technically sound, and do the data support the conclusions?

Reviewer #3: Yes

Reviewer #5: Yes

3. Has the statistical analysis been performed appropriately and rigorously? 

Reviewer #3: Yes

Reviewer #5: Yes

4. Have the authors made all data underlying the findings in their manuscript fully available?

Reviewer #3: Yes

Reviewer #5: Yes

5. Is the manuscript presented in an intelligible fashion and written in standard English?

Reviewer #3: Yes

Reviewer #5: Yes

6. Review Comments to the Author

Reviewer #3: I believe the revisions provided by the authors have addressed my comments. I support the acceptance of the manuscript in its current form.

Reviewer #5: In this work, the authors have done a rigorous job examining possible relationship between different types of qMRI and dMRI to explore the histology of brain tissues. All of the issues mentioned by the other reviewers have been addressed. It is interesting that there is a strong relationship between R1 and MTSat across tissue types, nevertheless, the diffusion metrics (FA, MD) are more correlated to R1 in GM than the WM. R1 is showed to be related to Myelin, which is primarily WM. The paper gives a sufficient explanation for this, but I would suggest also looking to radial diffusivity in their future work to see if the diffusion correlation is stronger there.

7. PLOS authors have the option to publish the peer review history of their article (what does this mean? ). If published, this will include your full peer review and any attached files.

**Do you want your identity to be public for this peer review?** For information about this choice, including consent withdrawal, please see our Privacy Policy .

Reviewer #3: No

Reviewer #5: No

---

## [Editor Report · Acceptance letter]

PONE-D-24-02122R2

PLOS ONE

Dear Dr. Steele,

I'm pleased to inform you that your manuscript has been deemed suitable for publication in PLOS ONE. Congratulations! Your manuscript is now being handed over to our production team.

Kind regards,

on behalf of

Dr. Md Nasir Uddin

Academic Editor

PLOS ONE